# WebRL: Training LLM Web Agents via Self-Evolving Online Curriculum Reinforcement Learning

**Zehan Qi**[1*], **Xiao Liu**[12*], **Iat Long Iong**[1], **Hanyu Lai**[1], **Xueqiao Sun**[1], **Jiadai Sun**[2]
**Xinyue Yang**[2], **Yu Yang**[2], **Shuntian Yao**[2], **Wei Xu**[13], **Jie Tang**[1], **Yuxiao Dong**[1]

[1]Tsinghua University      [2]Zhipu AI      [3]Shanghai Qi Zhi Institute

## ABSTRACT

Large language models (LLMs) have shown remarkable potential as autonomous agents, particularly in web-based tasks. However, existing LLM web agents face significant limitations: high-performing agents rely on expensive proprietary LLM APIs, while open LLMs lack the necessary decision-making capabilities. This paper introduces WebRL, a novel self-evolving online curriculum reinforcement learning framework designed to train high-performance web agents using open LLMs. Our approach addresses key challenges in this domain, including the scarcity of training tasks, sparse feedback signals, and policy distribution drift in online learning. WebRL incorporates a self-evolving curriculum that generates new tasks from unsuccessful attempts, a robust outcome-supervised reward model (ORM), and adaptive reinforcement learning strategies to ensure consistent improvement. We apply WebRL to transform Llama-3.1 models into proficient web agents, achieving remarkable results on the WebArena-Lite benchmark. Our Llama-3.1-8B agent improves from an initial 4.8% success rate to 42.4%, while the Llama-3.1-70B agent achieves a 47.3% success rate across five diverse websites. These results surpass the performance of GPT-4-Turbo (17.6%) by over 160% relatively and significantly outperform previous state-of-the-art web agents trained on open LLMs (AutoWebGLM, 18.2%). Our findings demonstrate WebRL's effectiveness in bridging the gap between open and proprietary LLM-based web agents, paving the way for more accessible and powerful autonomous web interaction systems. Code, model, and data will be available at `https://github.com/THUDM/WebRL`.

## 1 INTRODUCTION

Large language models (LLMs) have exhibited not only superior comprehension of human language, commonsense reasoning, and knowledge acquisition, but also significant potential in complex planning and logical reasoning, indicating their promising trajectory towards serving as autonomous LLM agents (Liu et al., 2024b). A diverse array of applications for LLM agents has proliferated, encompassing domains such as code generation (Jimenez et al., 2024), database manipulation (Zhou et al., 2023; Gu et al., 2024), and graphical user interface (GUI) interaction (Yang et al., 2023; Rawles et al., 2024; Xie et al., 2024). Among these, web agents powered by LLMs (Deng et al., 2024; Zheng et al., 2024; Lai et al., 2024; Pan et al., 2024) have garnered particular attention due to their extensive application prospects and unique potential for fostering authentic autonomous intelligence within the digital ecosystem.

Notwithstanding these advancements, existing LLM web agents, regardless of their performance metrics or architectural paradigms, remain under-developed. High-performing LLM web agents predominantly rely on meticulously crafted prompts in conjunction with proprietary LLM APIs (e.g., OpenAI GPT-4) for web page comprehension and manipulation, which is both expensive and

---

*Equal contribution. Emails: `qzh23@mails.tsinghua.edu.cn`, `shawliu9@gmail.com`
Work done when ZQ interned at Zhipu AI. Corresponding author: YD.

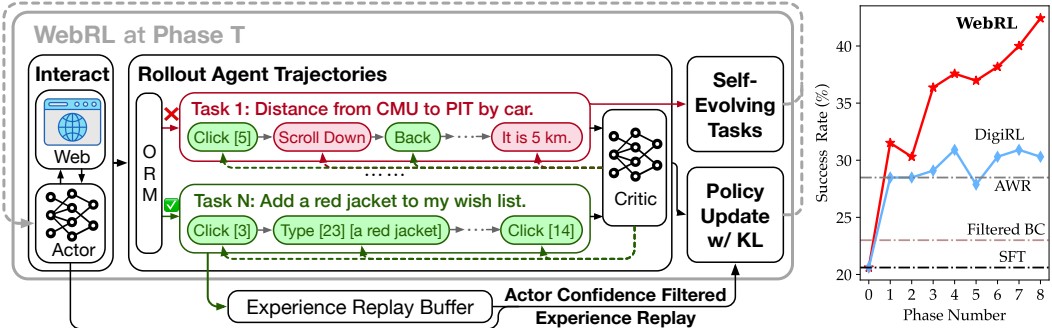

Figure 1: Overview of WEBRL. WEBRL is a self-evolving online curriculum reinforcement learning framework for LLM-based web agents, yielding consistent continual improvements throughout the iterative self-evolution.

time-intensive. Conversely, open-source LLMs exhibit notable deficiencies in their capability to function as proficient web agents, primarily due to the scarcity of decision-centric data in both pre-training and post-training periods. Despite recent endeavors (Lai et al., 2024; Pan et al., 2024) to train web agents on open LLMs via imitation learning, these approaches insufficiently leverage the inherently online nature of web interactions and fail to yield consistent, continual improvements.

**Challenges.** In this work, we propose to train high-performance web agents based on open LLMs within online environments, specifically utilizing WebArena (Zhou et al., 2024a). Our investigation has identified several critical challenges inherent to this task: 1) *Insufficiency of training tasks*: In contrast to offline datasets (Deng et al., 2024; Rawles et al., 2024) that facilitate agent training and evaluation on human-annotated oracle trajectories, online benchmarks such as WebArena (Zhou et al., 2024a) typically provide only a limited test set for evaluation purposes. This dearth of prede-fined training tasks significantly impedes the effective training of agents within these environments. 2) *Sparsity and cost of feedback signals*: The assessment of success for arbitrary web browsing tasks is difficult in the absence of task-specific evaluation functions. Moreover, unlike tasks in certain GUI datasets (e.g., AITW (Rawles et al., 2024) and WebShop (Yao et al., 2022)), those in WebArena are typically of long horizons, with oracle solutions averaging 10 steps. This characteristic introduces substantial sparsity in the available signals during online exploration. 3) *Policy distribution drift in online learning*: The absence of a predefined training set necessitates online exploration, inevitably leading to distribution drift in the agent's policy. This phenomenon is likely to induce catastrophic forgetting and performance degradation over time.

**Contribution: The WEBRL framework.** In response to these challenges, we introduce WEBRL, a self-evolving online curriculum reinforcement learning framework designed for training LLM web agents. To the best of our knowledge, this represents the first systematic framework enabling effective reinforcement learning for LLM web agents from initialization in online web environments. Through the application of WEBRL, we have successfully transformed a Llama-3.1-8B model into a proficient LLM web agent, elevating its success rate (SR) on WebArena-Lite (Zhou et al., 2024a; Liu et al., 2024c) from an initial 4.8% to 42.4% across a diverse set of five websites. Furthermore, when applied to Llama-3.1-70B, we achieve a remarkable 47.3% SR, surpassing the performance of the most advanced proprietary LLM API (GPT-4-Turbo, 17.6% SR) by over relatively 160% and significantly outperforming previous state-of-the-art web agents trained on open-source LLMs (AutoWebGLM (Lai et al., 2024), 18.2% SR).

The substantial performance gains from WEBRL can be attributed to several key architectural de-signs. To address the scarcity of web agent training tasks, we have devised a self-evolving online curriculum that harnesses the trial-and-error process inherent in exploration. This curriculum is un-derpinned by a robust outcome-supervised reward model (ORM) that we have newly developed. In each training phase, novel tasks are autonomously generated from unsuccessful attempts in the pre-ceding phase, facilitating a progressive learning trajectory. To mitigate the policy distribution shift induced by curriculum-based reinforcement learning, we incorporate a KL-divergence term between the reference and actor policies into our learning algorithm, thereby constraining policy updates and promoting stability. We implement an experience replay buffer augmented with a novel actor con-

fidence filtering strategy to ensure the fidelity of replayed experiences and prevent over-fitting to previously acquired knowledge.

In summary, our work makes the following significant contributions to the field:

- We introduce WEBRL, a novel self-evolving online curriculum RL framework for training LLM-based web agents. For the first time, it implements the infrastructure for RL in the WebArena environment, together with a strong ORM, to drive open LLMs to become capable web agents.
- WEBRL advances the RL for LLM agent training by addressing key challenges including the scarcity of training tasks, sparsity of feedback signals, and distribution drift in online learning. The self-evolving curriculum and adaptive learning strategies allow the consistent continual improvement of LLM web agents during iteration.
- We demonstrate WEBRL's substantial performance improvements over existing methodologies such as AWR and DigiRL, achieving state-of-the-art results on the WebArena-Lite benchmark. It surpasses the best proprietary LLM API and previously trained web agent on open LLMs by over 160% relatively. WEBRL has been adopted in AUTOGLM (Liu et al., 2024a), a series of phone use and browser use foundation GUI agents.

## 2 WEBRL: SELF-EVOLVING ONLINE CURRICULUM REINFORCEMENT LEARNING FOR LLM AGENTS ON WEB

We present a self-evolving online curriculum learning framework designed for training web agents, targeting the WebArena (Zhou et al., 2024a) environment. In this system, the agent continuously interacts with its environment to collect real-time trajectory data. We train an outcome-supervised reward model (ORM) to assess task success and use a KL-constrained policy update algorithm to prevent severe policy distribution drift during curriculum learning. A replay buffer is employed to retain knowledge and mitigate catastrophic forgetting. Furthermore, a self-evolving strategy is integrated to dynamically generate tasks that align with the agent's evolving capabilities. This approach enables the agent to improve incrementally, progressively handling more complex tasks. The overall training process can be found in Algorithm 1.

**Problem Formulation.** We model the process of completing the web task as a finite-horizon Markov Decision Process (MDP), denoted by $(S, A, R, \mathcal{T})$. Given a user instruction $I$, the agent is required to complete the corresponding task. The state $s$ is defined as the HTML content of the current web page along with the history of previous actions. The agent receives a reward of 1 upon successful task completion, and 0 otherwise. In the finite-horizon setting, the trajectory ends either when the task is accomplished or when the maximum number of interactions $T$ is exceeded. To explain our method clearly, we introduce the following notation. The policy $\pi(\cdot|s_t, I)$ represents the distribution over actions given the state $s_t$ and the instruction $I$. The value function $V(s_h, I) = \mathbb{E}_\pi \left[ \sum_{t=h}^{T} r(s_t, a_t, I) \right]$ represents the expected cumulative reward from the state $s_h$ under policy $\pi$. The action-value function $Q(s_t, a_t, I)$ is the expected cumulative reward for taking action $a_t$ on state $s_t$ and following policy $\pi$ thereafter: $Q(s_t, a_t, I) = r(s_t, a_t) + V(s_{t+1}, I)$.

**ORM Training.** In the curriculum learning process, we need to determine whether the corresponding instruction is completed based on the trajectory generated by the agent. Due to the lack of feedback from the environment, we train an LLM as the outcome-supervised reward model $\mathcal{M}_{\text{ORM}}$ to achieve this task success evaluation. $\mathcal{M}_{\text{ORM}}$ is utilized to assess whether the agent's rollout trajectory accomplishes a given task, providing a binary reward signal (0 for failure and 1 for success).

Similar to the approach in (Zhang et al., 2024e), we configure $\mathcal{M}_{\text{ORM}}$ to output "YES" or "NO" to indicate whether a trajectory successfully completes a task, leveraging the learned knowledge from the language head of $\mathcal{M}_{\text{ORM}}$. Given the limited context window of LLMs and the typically long length of HTML documents, we adopt a strategy akin to (Pan et al., 2024), keeping the HTML of only the final state to the input. In addition, the historical actions of agents, which provide information about previous steps are also included. Thus, the input to the model consists of: the instruction $I$, historical actions, and HTML of the final state. We wrap these components into the prompt asking the model to determine whether the trajectory successfully completes the task described by instruction $I$. To obtain the outcome, we compare the probabilities of generating "YES" and "NO" from

$\mathcal{M}_{\text{ORM}}$. If the probability of generating "YES" is higher than that of generating "NO", the task is considered successful, and the reward is set to 1. Otherwise, the reward is set to 0.

## 2.1 SELF-EVOLVING NEW INSTRUCTION FOR CURRICULUM LEARNING

A typical challenge in training LLM web agents within WebArena is the scarcity of training tasks, resonating with the situation of developing real-world web agents. Although the recent work (Liu et al., 2024c) has curated a trajectory fine-tuning set for WebArena, it only contains around 1k instructions with oracle trajectories, far from enough for training strong LLM web agents. To address this limitation and drive continuous improvement, we employ a self-evolving curriculum learning strategy. This method generates new training instructions at each phase. As the phase progresses, the generated instructions become increasingly complex, allowing the agent's capabilities to improve gradually. We implement a two-step process of generation and filtering, to produce tasks that are incrementally more challenging, while still being suitable for the agent's current capability. During the generation step, we use the in-breadth evolving approach (Xu et al., 2023) to create new instructions. We select instructions the model failed to complete in previous interaction phases as seeds for generating new instructions. Detailed prompts are provided in the Appendix § D. To ensure that the generated instructions are both feasible in the target environment and aligned with the desired difficulty level, we first filter them using the trained critic. Specifically, we use the critic to evaluate each new instruction by considering its initial state. We select instructions with critic scores between 0.05 and 0.75, ensuring that only tasks meeting our difficulty criteria are retained. We manually review generated tasks and identify tasks that cannot be completed in WebArena. Based on these findings, we develop a prompt (Figure 20) and use GPT-4o to exclude infeasible tasks in WebArena automatically. The resulting set of instructions is used for interaction and training in this phase.

## 2.2 REINFORCEMENT LEARNING FOR LLMS IN ONLINE WEB ENVIRONMENTS

In each phase of curriculum learning, the model progressively encounters and learns a new set of tasks. Considering this setting, a major challenge here is to avoid excessive policy distribution drift during each learning phase, which could lead to the catastrophic forgetting of previously acquired knowledge. Traditional approaches typically mitigate the issue by mixing data from different phases. However, in web agent tasks, intermediate steps do not receive direct process rewards, with only weak signals from the outcome of the final state. Consequently, even if an intermediate step is executed correctly, an error in later steps can easily lead to the final failure, resulting in misjudgment of the intermediate step and making it difficult to be reused. As a result, in this work, we primarily seek algorithmic improvements to address policy distribution drift more directly.

A potential solution comes from ideas in reinforcement learning from human feedback (RLHF) (Ouyang et al., 2022), where the Kullback-Leibler (KL) divergence between two policies is constrained to mitigate policy distribution drift. By adapting this to our curriculum learning setup, we aim to ensure that the policy in the current phase does not deviate too much from the policy in the previous phase, while still optimizing performance on new tasks. Let the policy from the previous phase be denoted as $\pi_{\text{ref}}$, and the current policy being optimized as $\pi_\theta$. The instruction distribution for the current phase is represented as $\rho(I)$. The objective for optimizing $\pi_\theta$ in the current phase can then be written as follows:

$$\max_{\pi_\theta} \mathbb{E}_{I \sim \rho(I), a_t \sim \pi_\theta(\cdot | s_t)} \left[ \sum_{t=0}^{T} \left( r(s_t, a_t, I) + \beta \log \pi_{\text{ref}}(a_t | s_t, I) \right) + \beta \mathcal{H}(\pi_\theta) \right] \quad (1)$$

where $\beta$ is a coefficient controlling the strength of the KL divergence constraint and $\mathcal{H}(\pi_\theta)$ represents the entropy of the current policy.

Following the work of (Rafailov et al., 2024a), the objective of eq. 1 can be interpreted as a maximum entropy reinforcement learning problem. The optimal policy $\pi^*$ for this problem can be expressed as:

$$\pi^*(a_t | s_t, I) = e^{(Q^*(s_t, a_t, I) - V^*(s_t, I)) / \beta} \quad (2)$$

where $V^*(s_t, I)$ is the optimal value function, representing the expected cumulative reward under the optimal policy $\pi^*$. $Q^*(s_t, a_t, I)$ is the optimal action-value function. The relationship between

$Q^*$ and $V^*$ is given by:

$$Q^*(s_t, a_t, I) = \begin{cases} r(s_t, a_t, I) + \beta \log \pi_{\text{ref}}(a_t|s_t, I) + V^*(s_{t+1}, I), & \text{if } s_{t+1} \text{ is not terminal} \\ r(s_t, a_t, I) + \beta \log \pi_{\text{ref}}(a_t|s_t, I), & \text{if } s_{t+1} \text{ is terminal} \end{cases} \quad (3)$$

Based on eq. 2 and eq. 3, we can derive:

$$\beta \log \frac{\pi^*(a_t|s_t, I)}{\pi_{\text{ref}}(a_t|s_t, I)} = r(s_t, a_t, I) + V^*(s_{t+1}, I) - V^*(s_t, I) = A^*(s_t, a_t, I) \quad (4)$$

Here, $A^*(s_t, a_t, I)$ indicates the advantage of taking action $a_t$ in state $s_t$ compared to the average reward expected in that state. Based on the condition, we can formulate the loss function of policy $\pi_\theta$ as:

$$\mathcal{L}(\pi_\theta) = \mathbb{E}_\nu \left[ \left( \beta \log \frac{\pi_\theta(a|s, I)}{\pi_{\text{ref}}(a|s, I)} - A^*(s, a, I) \right)^2 \right] \quad (5)$$

where $\nu(s)$ represents the distribution of experience used for training. Note that our algorithm operates in the off-policy manner. A comprehensive derivation, thorough analysis, and a detailed comparison with other RL algorithms can be found in Appendix A.

**What does the update do?** To gain a mechanistic understanding of the loss function, we analyze the gradient of the loss function, $\mathcal{L}(\pi_\theta)$. The gradient with respect to the parameters $\theta$ can be expressed as:

$$\nabla_\theta \mathcal{L}(\pi_\theta) = -2\beta \mathbb{E}_\nu \left[ \left( \underbrace{A^*(s, a, I)}_{\text{update direction}} - \underbrace{\beta \log \frac{\pi_\theta(a|s, I)}{\pi_{\text{ref}}(a|s, I)}}_{\text{KL divergence constraint}} \right) \underbrace{\nabla_\theta \log \pi_\theta(a|s, I)}_{\text{sft loss}} \right] \quad (6)$$

The gradient demonstrate the following attributions:

- When the advantage $A^*(s, a, I) > 0$, action $a$ is valuable, so its probability should increase. If $\pi_\theta$ is lower than $\pi_{\text{ref}}$, this increase will be amplified, especially as the gap between them grows. If $\pi_\theta$ is already higher than $\pi_{\text{ref}}$, the increase will be moderated to avoid excessive deviation.
- When $A^*(s, a, I) < 0$, the action is suboptimal, so its probability should decrease. If $\pi_\theta$ is lower than $\pi_{\text{ref}}$, the KL divergence constraint will limit how much it can be reduced to avoid a large divergence. If $\pi_\theta$ is higher than $\pi_{\text{ref}}$, a larger decrease will be allowed.
- The parameter $\beta$ controls the strength of the KL divergence constraint. Adjusting $\beta$ can help fine-tune this constraint. For instance, increasing $\beta$ can prevent unnecessary boosts in action probabilities when $\pi_{\text{ref}}$ already assigns a high probability to an action.

**Training a Reliable Advantage Estimator.** A reliable advantage estimator is essential for effective policy updates. We train a value network $V(s_t, I)$ and use Generalized Advantage Estimation (GAE) (Schulman et al., 2015) to compute the advantage. In our setting, we only receive a binary reward (0 or 1) at the final step, with no intermediate rewards (*i.e.*, intermediate rewards are effectively zero). Following recent approaches (Farebrother et al., 2024), we train the value network using a cross-entropy objective. The loss function for the value network $V$ is defined as:

$$\mathcal{L}(V) = -\mathbb{E}_\nu \left[ r(s_T, a_T, I) \log V(s, a, I) + (1 - r(s_T, a_T, I)) \log(1 - V(s, a, I)) \right] \quad (7)$$

In line with (Bai et al., 2024), we focus solely on the next-step and final-step advantage estimators, since there is no intermediate reward.

$$A(s_t, a_t, I) = \lambda \big( r(s_t, a_t, I) + V(s_{t+1}, I) - V(s_t, I) \big) + (1 - \lambda) \big( r(s_T, a_T, I) - V(s_t, I) \big) \quad (8)$$

where $\lambda$ is a balancing factor that controls the trade-off between bias and variance in advantage estimation. We set $\lambda$ as 0.5 in our work.

**Experience Replay Buffer with Actor Confidence Filtering.** In addition to controlling the policy distribution drift at the algorithmic level through KL, we also implement an adaptive replay buffer to alleviate knowledge forgetting at the data level. Specifically, we only store those successful trajectories (which can be sparse) from each phase in the replay buffer. During phase $i$, we use the actor from the last phase to compute the perplexity of all actions in the buffer. Actions with a

perplexity within the range of 1/0.95 to 1/0.5, along with their corresponding states, are added to the training data for the current phase. This filtering process excludes both over-familiar data and data that remains too challenging for the actor. Additionally, by storing only successful trajectories, we avoid the challenge of accurately estimating intermediate states for incorrect trajectories from previous phases.

## 3 EXPERIMENTS

### 3.1 ENVIRONMENTS AND BASELINES

**Environments.** The effectiveness of our method and the baseline models is evaluated using the WebArena environment (Zhou et al., 2024a). WebArena is particularly well-suited to our needs, as it provides a highly interactive platform that supports online learning. Additionally, WebArena encompasses a variety of websites, including OpenStreetMap (Map), Reddit, GitLab, online store content management system (CMS), and OneStopShop (OSS), making it an ideal benchmark for comprehensively assessing model performance on web tasks. In the original WebArena environment, a total of 812 instructions are provided. Considering the cost of testing, we use 165 test cases from WebArena-Lite (Liu et al., 2024c) for evaluation.

**Baselines.** We compare WEBRL with several methods utilizing prompting techniques, as well as models trained with alternative methods. For proprietary models, we select GPT-4-Turbo-2024-0409 (GPT-4-Turbo) (Achiam et al., 2023) and GPT-4o. In addition to AWM (Wang et al., 2024b) and WebPilot (Zhang et al., 2024f), we also use the results of models under the simple prompt as baselines. Details of the simple prompt can be seen in Appendix § D. For the open-source models, we train Llama3.1 (Dubey et al., 2024) using various approaches as baselines. Specifically, we employ imitation learning, also referred to as supervised fine-tuning (SFT), to train both the Llama3.1-8B and Llama3.1-70B. The training data is derived from publicly available human-labeled demonstrations, sourced from the WebArena-Lite. In addition, we also explore several reinforcement learning methods for comparison, including Filtered Behavior Cloning (Filtered BC) (Pan et al., 2024), advantage-weighted regression (AWR) (Peng et al., 2019) and DigiRL (Bai et al., 2024). For WEBRL and the reinforcement learning-based baselines, we utilize SFT-trained Llama3.1-8B as the initial model for the actor. For the critic, we employ Llama3.1-8B, augmented with a randomly initialized value head to serve as the initial model. The training details of WEBRL and baselines can be found in Appendix § B.

**ORM.** WebArena-Lite (Liu et al., 2024c) provides training samples along with a corresponding reward function. We further enhance this set of data by introducing task rewrites, as well as modifying certain data variables, such as place names and product names. We also make adjustments to the associated reward function. $\mathcal{M}_{ORM}$ is trained using rollouts of WEBRL and part of baseline methods on this set of tasks, with evaluation results determined by the reward function.

### 3.2 MAIN RESULTS

Our main results, presented in Table 1, show that Llama3.1-8B trained using WEBRL achieves an average accuracy of 42.4%, outperforming all baselines, including prompting and training alternatives. Notably, WEBRL excels in specific tasks such as Gitlab (46.7%) and CMS (54.3%), demonstrating its ability to address complex web tasks effectively. Reinforcement learning-based approaches outperform those based on imitation learning, such as SFT and Filtered BC, which tend to over-repeat certain actions. For instance, in CMS, SFT models often over-optimize "Scroll Down" actions. This over-optimize can cause the model to become trapped in local loops, thereby hindering its ability to achieve the overall task objective effectively. In contrast, reinforcement learning mitigates this by using a critic to estimate the value of each step, optimizing for long-term rewards, hence enabling better handling of complex, multi-step tasks. WEBRL also consistently outperforms DigiRL. A key limitation of DigiRL is that it performs policy updates on a fixed set of tasks, which can lead to the model learning only a subset of operations within these tasks due to sparse feedback. This may cause the model to converge to suboptimal solutions and limit its ability to fully explore its potential. WEBRL addresses this by employing self-evolving curriculum learning, adjusting task complexity based on the model's abilities, and promoting broader exploration and continuous improvement.

Table 1: Task success rate (SR) of WEBRL and other comparison methods, evaluated on WebArena-Lite (Zhou et al., 2024a; Liu et al., 2024c), a human-verified subset of WebArena (* denotes results on full WebArena taken from literature reporting). The **best** and second-best models are highlighted.

| Models | #Params | Reddit | Gitlab | CMS | Map | OSS | Avg. SR |
|---|---|---|---|---|---|---|---|
| *Prompting Proprietary LLMs* | | | | | | | |
| GPT-4-Turbo | N/A | 10.5 | 16.7 | 14.3 | 36.7 | 13.3 | 17.6 |
| GPT-4o | N/A | 10.5 | 10.0 | 20.0 | 20.0 | 11.1 | 13.9 |
| Llama3.1-Instruct (Dubey et al., 2024) | 8B | 0.0 | 3.3 | 2.9 | 3.3 | 11.1 | 4.8 |
| Llama3.1-Instruct (Dubey et al., 2024) | 70B | 10.5 | 16.7 | 17.1 | 20.0 | 4.4 | 12.7 |
| AWM +GPT-4-0613* (Wang et al., 2024b) | N/A | 50.9 | 31.8 | 29.1 | **43.3** | 30.8 | 35.5 |
| WebPilot + GPT-4o* (Zhang et al., 2024f) | N/A | 65.1 | 39.4 | 24.7 | 33.9 | 36.9 | 37.2 |
| *Training Open LLMs* | | | | | | | |
| AutoWebGLM (Lai et al., 2024) | 6B | 9.4 | 15.0 | 28.6 | 24.8 | 17.1 | 18.2 |
| GLM-4-Chat (GLM et al., 2024) | 9B | 5.3 | 10.0 | 6.7 | 3.3 | 6.7 | 6.1 |
| GLM-4 + SFT (BC) | 9B | 47.4 | 13.3 | 31.4 | 23.3 | 13.3 | 22.4 |
| GLM-4 + Filtered BC | 9B | 52.6 | 10.0 | 31.4 | 26.7 | 20.0 | 24.8 |
| GLM-4 + AWR (Peng et al., 2019) | 9B | 52.6 | 16.7 | 34.3 | 30.0 | 22.2 | 27.9 |
| GLM-4 + DigiRL (Bai et al., 2024) | 9B | 63.2 | 30.0 | 34.3 | 26.7 | 26.7 | 31.5 |
| GLM-4 + WEBRL (ours) | 9B | 57.9 | **50.0** | 48.6 | 36.7 | 37.8 | 43.0 |
| Llama3.1 + SFT (BC) | 8B | 36.8 | 6.7 | 20.0 | 33.3 | 17.8 | 20.6 |
| Llama3.1 + Filtered BC | 8B | 52.6 | 20.0 | 31.4 | 23.3 | 8.9 | 23.0 |
| Llama3.1 + AWR (Peng et al., 2019) | 8B | 57.9 | 26.7 | 31.4 | 26.7 | 17.8 | 28.5 |
| Llama3.1 + DigiRL (Bai et al., 2024) | 8B | 57.9 | 26.7 | 37.1 | 33.3 | 17.8 | 30.3 |
| Llama3.1 + WEBRL (ours) | 8B | 63.2 | 46.7 | **54.3** | 36.7 | 31.1 | 42.4 |
| Llama3.1-Instruct (Dubey et al., 2024) | 70B | 10.5 | 16.7 | 17.1 | 20.0 | 4.4 | 12.7 |
| Llama3.1 + SFT (BC) | 70B | 52.6 | 20.0 | 20.0 | 26.7 | 13.3 | 23.0 |
| Llama3.1 + WEBRL (ours) | 70B | **78.9** | **50.0** | **54.3** | 40.0 | **44.4** | **49.1** |

## 3.3 SCALING EFFECT OF WEBRL

We further validate the effectiveness of WEBRL on larger-scale models by training Llama3.1-70B using WEBRL. Due to cost constraints, we only conduct 4 phases of curriculum learning. The specific results are presented in Table 1. After training with WEBRL, Llama3.1-70B achieves an overall accuracy of 47.3%, reflecting a 24.3% improvement over the accuracy achieved with SFT. This indicates that WEBRL is scalable and can be effectively applied to larger-scale models. Furthermore, when comparing the performance improvement from Llama3.1-8B to Llama3.1-70B achieved through SFT, WEBRL demonstrates even greater performance gains as the model scale increases.

## 3.4 DISTRIBUTION ANALYSIS OF ERROR TYPES

We compare the performance of Llama 3.1-8B trained with WEBRL against baseline methods across different error types: "Fail to Recover", "Get Stuck Midway", "Stop at Wrong Page", and "Fail to Make Reasonable Attempt", as shown in Figure 2. WEBRL demonstrates significant advantages in reducing the "Get Stuck Midway" error, especially compared to SFT and Filtered BC. The "Get Stuck Midway" error typically arises when the model gets trapped in a loop, repeatedly executing the same action without making progress. Reinforcement learning helps mitigate this issue by optimizing each action while considering its overall impact on the task, enabling the model to make more effective decisions. Additionally, models trained with WEBRL demonstrate enhanced robustness in handling the "Fail to Recover" error. Through curriculum learning, the model gradually learns how to adapt its actions when encountering failures. For example, if the search query "Pharmacy near CMU within a 20-minute walking distance" does not yield the desired results, the model learns to modify the query to "Pharmacy near CMU" and attempts the search again, rather than repeating ineffective actions. In addition, WEBRL exhibits the lowest error rate on both "Stop at Wrong Page" and "Fail to Make Reasonable Attempt" errors, indicating the model trained with WEBRL has a more profound comprehension of the relationship between tasks and web pages. It can better identify the correct page needed to complete a specific task, reducing the chances of mistakenly stopping on the wrong page or navigating to an incorrect page.

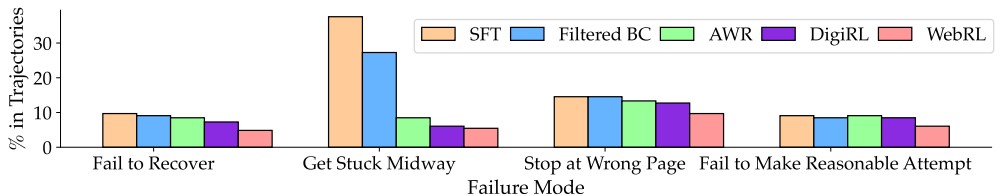

Figure 2: Distribution analysis of error types for WEBRL and baseline methods.

## 3.5 PERFORMANCE ON TASKS WITH VARYING COMPLEXITY

We further analyze the performance of WEBRL and baselines across instructions of varying complexity, as shown in Figure 3. Instruction complexity is measured by the number of requirements in the task. For example, the instruction "What are the top-3 best-selling products in Jan 2023" has two requirements: identifying the top-3 products and specifying the timeframe, giving it a complexity level of 2. Our results show that WEBRL performs well across different complexity levels, particularly excelling in more complex instructions. In contrast, while DigiRL uses online learning, it struggles with higher complexity due to its focus on a fixed set of tasks, limiting its adaptability. This highlights the effectiveness of our self-evolving curriculum learning strategy, which progressively increases task complexity based on the model's capacity, enabling better performance on challenging tasks.

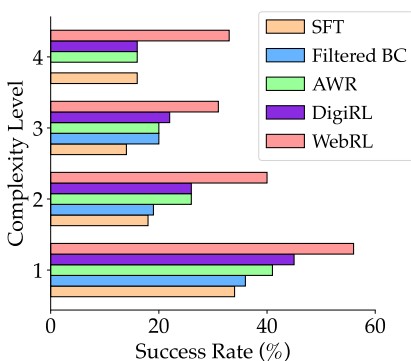

Figure 3: Accuracy of WEBRL and baselines for tasks with different complexity.

## 3.6 PERFORMANCE ON TASKS WITH VARYING STEP REQUIREMENTS

We evaluate the performance of Llama3.1-8B, trained using WEBRL and baseline methods, on tasks with varying step requirements. To determine the required step count for each task, we exclude any tasks that no model completes and use the trajectory with the fewest steps as the required step count for each remaining task. The results are shown in Figure 4. It can be seen that the performance of both the SFT and Filtered BC trained models shows a noticeable decline as the task length increases. This is likely because these models optimize individual steps without considering the cumulative impact, making them less effective on long-horizon tasks. DigiRL-trained model improves performance on medium-length tasks but struggles with longer tasks (more than 10 steps). This limitation may stem from DigiRL's online learning on a fixed set of tasks. Even when the model executes intermediate steps correctly, it doesn't receive positive rewards if errors occur in later steps, making it harder for the model to learn how to complete tasks that require many steps effectively. In contrast, WEBRL overcomes this issue with curriculum learning, progressively increasing task difficulty. This approach enhances the model's ability to handle long sequences, leading to significant performance improvements on tasks requiring long-term planning compared to other methods.

## 3.7 ABLATION STUDY

We conduct an ablation study to evaluate the impact of the replay buffer, KL-constrained policy update algorithm, and the curriculum learning strategy on WEBRL. To assess their contributions, we compare WEBRL with four alternative models: (1) WEBRL w/o replay buffer, where training uses only the current interaction trajectory, (2) WEBRL w/o KL, where the policy is updated using REINFORCE with value function baseline (the gradient is $\mathbb{E}_\nu[(A(s, a, I)\nabla_\theta \log \pi_\theta(a|s, I)])$ but retains the replay buffer, (3) WEBRL w/o KL & replay buffer, which uses neither a replay buffer nor the KL-constrained policy update algorithm, and (4) WEBRL w/o CL, which ablates the curriculum learning approach, utilizing only the instructions generated in the first phase. The results, shown in Figure 5, reveal that when the replay buffer is removed, both WEBRL w/o replay buffer and WEBRL w/o KL & replay buffer experience worsening performance over time. This decline occurs because

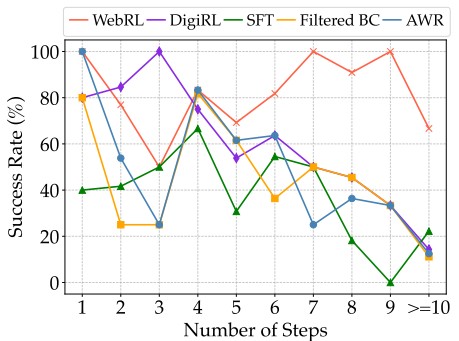

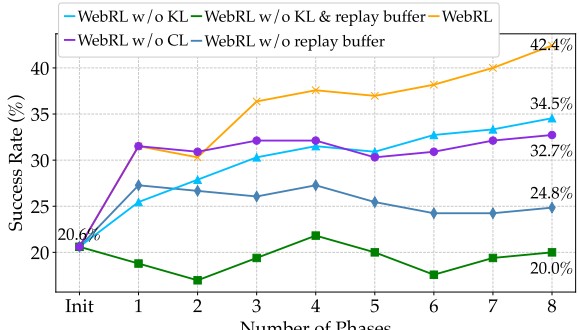

Figure 4: Accuracy of WEBRL and baselines for tasks requiring different steps.

Figure 5: Ablation study of WEBRL on replay buffer, KL-constrained policy update and curriculum strategy.

the models lose access to earlier experiences and focus only on recent data, leading to knowledge degradation. Comparing WEBRL and WEBRL w/o KL, WEBRL consistently performs better, due to the incorporation of KL-constrained policy update algorithm. When the replay buffer is not used, the KL-constrained policy update algorithm degrades more slowly than REINFORCE with a value function baseline. This is because the KL constraint better retains knowledge from past policy iterations by explicitly controlling the divergence between the current and previous policies. In contrast, REINFORCE with a value function baseline is more prone to overfitting the data from the current phase of training, leading to less stable performance over time. Overall, the KL-constrained policy update algorithm is more effective at balancing the retention of past knowledge with the learning of new information. When comparing WEBRL to WEBRL w/o CL, both exhibit an overall upward trend due to online learning. However, WEBRL w/o CL progresses more slowly and reaches a lower performance ceiling because it operates within a fixed task framework, whereas WEBRL generates new tasks that adapt to its evolving capabilities. This highlights the effectiveness of our self-evolving curriculum learning approach. Additional ablation studies to further investigate the effects of curriculum learning are presented in Figure 12.

**The impact of $\beta$.** We investigate the effect of $\beta$ on performance with and without the replay buffer, as shown in Figure 6. The study involves curriculum learning in a single phase. First, when $\beta$ is too small (e.g., $\beta = 0.01$), performance declines regardless of the use of the replay buffer. This happens because a small $\beta$ leads to weak control over KL divergence, causing the model to overfit the current data. Second, without the replay buffer, performance initially improves as $\beta$ increases but declines when $\beta$ becomes too large, indicating that larger $\beta$ (e.g., $\beta \geq 1$) will overly restrict KL divergence, limiting the model's ability to update its policy and learn effectively. In contrast, with the replay buffer, performance stays high

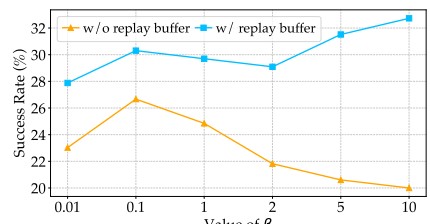

Figure 6: The impact of $\beta$ of KL-constrained policy update algorithm on the model's performance.

even at larger $\beta$ values. The replay buffer's historical experiences allow for more frequent parameter updates, supporting steady improvement despite higher $\beta$.

**The influence of perplexity.** We analyze the impact of using perplexity to select data from replay buffer for training. Various perplexity thresholds are tested in the first learning phase, and the results are summarized in Table 2. It can be observed that training on data with very low perplexity (range [1, 10/9]) leads to performance deterioration. This suggests that repeatedly learning overly famil-

Table 2: The impact of perplexity in replay buffer filtering of WEBRL.

| $[\mathbf{1},\infty]$ | $[\mathbf{1},\frac{\mathbf{10}}{\mathbf{9}}]$ | $[\frac{\mathbf{10}}{\mathbf{9}},\frac{\mathbf{10}}{\mathbf{5}}]$ | $[\frac{\mathbf{10}}{\mathbf{5}},\infty]$ |
|---|---|---|---|
| 29.1 | 28.5 | 30.9 | 23.0 |

iar data harms the model. Similarly, training exclusively on data with high perplexity (above 10/5) also degrades performance, likely due to the model struggling with unfamiliar data, causing a significant shift in policy distribution and hindering generalization. Optimal performance is achieved when training on data with a perplexity range of [10/9, 10/5], indicating that a balance between simple and complex data enhances model performance by focusing on moderately difficult examples.

Table 3: Evaluation on output-supervised methods (baselines adopted from (Pan et al., 2024)). Our ORM, without accessing proprietary GPT-4, performs the best among all.

|  | Our ORM (8B) | GPT-4 | Captioner + GPT-4 | GPT-4V |
|---|---|---|---|---|
| **Test Dataset (%)** | 80.8 | 71.9 | 72.6 | 71.2 |
| **Rollout (%)** | 79.4 | 71.2 | 73.3 | 70.5 |

## 3.8 EVALUATION OF ORM

In the WEBRL framework, continuous improvement relies heavily on the performance of ORM, which plays a crucial role in evaluating interaction trajectories to guide the agent's learning. To assess ORM's effectiveness, we compare its performance with several baseline models, including GPT-4-Turbo using identical inputs of our ORM, Captioner + GPT-4-Turbo, and GPT-4V, both using the same prompts with Pan et al. (2024). We evaluate ORM and the baselines on two datasets: the WebArena-Lite test set and 100 sampled rollouts which are manually labeled. For the WebArena-Lite test data, we use its reward function outputs as labels. The results, shown in Table 3, indicate that while the baseline models achieve a consistent accuracy just above 70%, our ORM outperforms them with approximately 80% accuracy.

## 4 RELATED WORKS

**Adopting LLMs as Agent.** As LLMs advance, they extend beyond text generation (Zheng et al., 2023; Zhao et al., 2023) and reasoning (Zelikman et al., 2024; Zhang et al., 2024d) to device control. Research follows training-free approaches that enhance LLMs via prompting (Yan et al., 2023; He et al., 2024; Zhang et al., 2024c) and system design (Liu et al., 2023; Yang et al., 2023; Wang et al., 2024a; Wu et al., 2024; Iong et al., 2024; Zhang et al., 2024a), but these remain limited by underlying LLM capabilities (Chen et al., 2023; Zeng et al., 2023; Xie et al., 2024). Training-based methods using imitation learning require costly expert demonstrations (Zhang & Zhang, 2023; Gur et al., 2024; Deng et al., 2024; Hong et al., 2024; Rawles et al., 2024; Zhang et al., 2024b). While some use GPT-4 for demonstration generation (Chen et al., 2023), they often focus on individual actions without considering long-term effects, limiting generalization (Ghosh et al., 2021; Bai et al., 2024). Some approaches use sampling to estimate long-term effects (Lai et al., 2024; Putta et al., 2024), while others leverage reinforcement learning (Carta et al., 2023; Bai et al., 2024; Pan et al., 2024; Tan et al., 2024; Zhai et al., 2024). Most rely on static task sets, hindering continuous improvement. We propose a dynamic task generation framework with a policy-update algorithm for ongoing enhancement.

**Reinforcement Learning for LLMs.** RL has gained traction in LLM training for preference optimization (Ouyang et al., 2022; Casper et al., 2024) and reasoning (Hao et al., 2023; Pang et al., 2024). For device control tasks requiring sequential decision-making, existing research uses online learning methods like AgentQ (Putta et al., 2024) with DPO (Rafailov et al., 2024b), or actor-critic architectures (Bai et al., 2024; Zhou et al., 2024b; Zhai et al., 2024), which we adopt. However, web tasks often provide only binary success/failure feedback after multiple interactions, potentially penalizing correct intermediate actions. Current research typically focuses on fixed task sets, limiting continuous improvement. We propose an autonomous curriculum learning mechanism that dynamically generates tasks based on agent skills, plus a KL-constrained policy update algorithm and specialized replay buffer to reuse historical data and prevent knowledge forgetting during curriculum updates.

## 5 CONCLUSION

In this work, we introduce WEBRL, a novel self-evolving online curriculum reinforcement learning framework for training LLM-based web agents. By addressing key challenges such as the scarcity of training tasks, feedback signal sparsity, and policy distribution drift, WEBRL enables continual and consistent improvement in agent performance within online environments like WebArena. Our approach demonstrates substantial performance gains, significantly surpassing existing state-of-the-art web agents and proprietary LLM APIs. These results highlight the effectiveness of WEBRL in advancing the capabilities of open-source LLMs for web-based tasks.

ACKNOWLEDGEMENT

We would like to thank the anonymous reviewers for their suggestions in refining this work. This work is in part supported by NSFC 62495063, 624B2084, 62425601, and Tsinghua University (Department of Computer Science and Technology)-Siemens Ltd., China Joint Research Center for Industrial Intelligence and Internet of Things (JCIIOT). Wei Xu is supported by the National Key R&D Program of China 2023YFC3304802 and NSFC under Grant U2268202 and 62176135.

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

# A  DETAILS OF POLICY UPDATE ALGORITHM IN WEBRL

## A.1  DERIVATION

By substituting eq. 3 into eq. 2, we can obtain:

$$\beta \log \frac{\pi^*(a_t|s_t, I)}{\pi_{\text{ref}}(a_t|s_t, I)} = r(s_t, a_t, I) + V^*(s_{t+1}, I) - V^*(s_t) = A^*(s_t, a_t, I) \tag{9}$$

$$\pi^*(a_t|s_t, I) = \pi_{\text{ref}}(a_t|s_t, I) \exp\left(\frac{1}{\beta} A^*(s_t, a_t, I)\right) \tag{10}$$

$\pi^*$ is the optimal solution to our target (eq. 1). If $\pi_\theta$ is represented by a function approximator, we aim to make it as close as possible to $\pi^*$.

$$\arg\min_\theta \mathbb{E}_\nu \left[ (\log \pi_\theta(a|s, I) - \log \pi^*(a|s, I))^2 \right]$$

$$= \arg\min_\theta \mathbb{E}_\nu \left[ \left( \log \pi_\theta(a|s, I) - \log \pi_{\text{ref}}(a|s, I) - \frac{1}{\beta} A^*(s, a, I) \right)^2 \right] \tag{11}$$

$$= \arg\min_\theta \mathbb{E}_\nu \left[ \left( \beta \log \frac{\pi_\theta(a|s, I)}{\pi_{\text{ref}}(a|s, I)} - A^*(s, a, I) \right)^2 \right]$$

Hence, the loss function can be defined as:

$$\mathcal{L}(\pi_\theta) = \mathbb{E}_\nu \left[ \left( \beta \log \frac{\pi_\theta(a|s, I)}{\pi_{\text{ref}}(a|s, I)} - A^*(s, a, I) \right)^2 \right] \tag{12}$$

where $\nu$ represents the distribution of experience used for training. For any given state-action pair, $\pi_\theta$ is expected to match the target policy $\pi^*(a|s, I)$. There is no restriction on the data distribution $\nu$, indicating the algorithm can function effectively in an off-policy setting. Since $A*$ is unknown, we estimate $A^*(s, a, I)$ using eq. 7 and eq. 8 to train the policy $\pi_\theta$. The feasibility of this approach is demonstrated in Appendix A.2.

**Further Analysis.** Although $\nu$ can follow any distribution, achieving stable policy improvement often requires some control over $\nu$. The primary goal is to enhance the probability of actions that successfully complete the task and to address the deficiencies of the current policy $\pi_\theta$. Therefore, $\nu$ typically consists of data sampled from the current policy being trained and prior successful experiences.

**Why not use KL divergence to measure the distance between $\pi^*$ and $\pi_\theta$.** For eq. 11, many studies use KL divergence to measure the distance between two policies. When KL divergence is used, the optimization goal of $\theta$ is:

$$\arg\min_\theta \mathbb{E}_{s \sim d(s)} \left[ D_{\text{KL}} \left( \pi^*(\cdot|s, I) || \pi_\theta(\cdot|s, I) \right) \right]$$

$$= \arg\max_\theta \mathbb{E}_{s \sim d(s)} \mathbb{E}_{a \sim \pi^*(a|s, I)} \left[ \log \pi_\theta(a|s, I) \right]$$

$$= \arg\max_\theta \mathbb{E}_{s \sim d(s)} \int_a \pi_{\text{ref}}(a|s, I) \exp(\frac{1}{\beta} A^*(s, a, I) \log \pi_\theta(a|s, I) \mathrm{d}a \tag{13}$$

$$= \arg\max_\theta \mathbb{E}_{s \sim d(s)} \mathbb{E}_{a \sim \pi_{\text{ref}}(a|s, I)} \left[ \log \pi_\theta(a|s, I) \exp(\frac{1}{\beta} A^*(s, a, I)) \right]$$

where $d(s)$ is a distribution of state. The choice of using eq. 11 instead of eq. 13 as the optimization target can be explained as follows: (1) Eq. 13 imposes stronger restrictions on the training data. Specifically, it requires the training data to conform to the distribution of $\pi_{\text{ref}}$.

When $\pi_{\text{ref}}$ fails to capture previous successful experiences, it becomes difficult to sample these experiences from $\pi_{\text{ref}}$, which in turn, can lead to further forgetting of those experiences. (2) Both eq. 11 and eq. 13 aim to approximate the distance between two distributions. The use of Mean Squared Error as a metric to measure the distance between policy distributions is also employed in Fujimoto & Gu (2021). (3) Eq. 13 is unable to decrease the probability of actions with negative advantage. When correct action is hard to sample from $\pi_{\text{ref}}$, eq. 13 may instead increase the probability of actions with negative advantage, compounding the challenge of sampling correct actions.

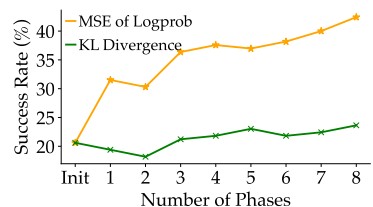

Figure 7: Comparison of using eq. 11 and eq. 13 as target.

We also conduct experiments to further validate the superiority of eq. 11, with the results presented in Figure 7. In these experiments, eq. 13 is trained exclusively on data sampled from $\pi_{\text{ref}}$, while eq. 11 utilize both replay buffer data and sampled data.

**When discount factor $\gamma$ is used.** When considering the discount factor, eq. 9 is modified as follows:

$$\beta \log \frac{\pi^*(a_t|s_t, I)}{\pi_{\text{ref}}(a_t|s_t, I)} = r(s_t, a_t, I) + \gamma V^*(s_{t+1}, I) - V^*(s_t) = A^*(s_t, a_t, I) \tag{14}$$

This will not affect the subsequent derivation of the policy loss function (eq. 12). The loss function for the critic remains unchanged. However, the equation to calculate the advantage is modified as follows:

$$A(s_t, a_t, I) = \lambda \big( r(s_t, a_t, I) + \gamma V(s_{t+1}, I) - V(s_t, I) \big) + (1-\lambda)\big( \gamma^{T-t} r(s_T, a_T, I) - V(s_t, I) \big) \tag{15}$$

In our experiment, we use the discount factor and set its value to 0.9.

We use eq. 7 and eq. 8 to estimate $A^*(s, a, I)$. The training of critic (eq. 7) also operates in an off-policy manner.

## A.2 PROOF

To prove the feasibility of replacing $A^*$ with an advantage estimator $A$ based on a value network $V$, which is trained on data collected throughout the training process, we aim to demonstrate that using $V$ to train the policy can lead to policy improvement.

The current policy is denoted by $\pi_{\text{old}}$. The training dataset $D$ is composed of trajectories sampled using $\pi_{\text{old}}$ as well as historical experiences stored in a replay buffer. The behavioral policy associated with the dataset $D$ is referred to as $\pi_\mu$. The corresponding value function and action-value function under $\pi_\mu$ are denoted as $V^{\pi_\mu}$ and $Q^{\pi_\mu}$, respectively. Below, we demonstrate that minimizing the objective $\arg\min_\theta \mathbb{E}_D[(\log_{\pi_\theta}(a_t|s_t) - \exp(\frac{Q^{\pi_\mu}(s_t,a_t) - V^{\pi_\mu}(s_t)}{\beta}))^2]$ can lead to policy improvement. For brevity, we omit $I$ in the following proof.

Let the new policy $\pi_{\text{new}}(a_t|s_t)$ be defined as:

$$\pi_{\text{new}}(a_t|s_t) = \exp\left( \frac{Q^{\pi_\mu}(s_t, a_t) - V^{\pi_\mu}(s_t)}{\beta} \right) \tag{16}$$

It can be shown that: $D_{\text{KL}}(\pi_{\text{new}}||\pi_{\text{new}}) \le D_{\text{KL}}(\pi_\mu||\pi_{\text{new}})$. From this, we derive the following inequality:

$$\mathbb{E}_{a_t \sim \pi_{\text{new}}}\left[ \log \pi_{\text{new}}(a_t|s_t) - \frac{Q^{\pi_\mu}(s_t, a_t)}{\beta} + \frac{V^{\pi_\mu}(s_t)}{\beta} \right]$$
$$\le \mathbb{E}_{a_t \sim \pi_\mu}\left[ \log \pi_\mu(a_t|s_t) - \frac{Q^{\pi_\mu}(s_t, a_t)}{\beta} + \frac{V^{\pi_\mu}(s_t)}{\beta} \right] \tag{17}$$

Since $V^{\pi_\mu}(s_t)$ depends only on the state $s_t$ and not on the actions, it can be factored out of the expectation. We can rewrite the inequality as:

$$\mathbb{E}_{a_t \sim \pi_{\text{new}}}\left[ Q^{\pi_\mu}(s_t, a_t) - \beta \log \pi_{\text{new}}(a_t|s_t) \right] \ge V^{\pi_\mu}(s_t) \tag{18}$$

where $V^{\pi_\mu}(s_t) = \mathbb{E}_{a \sim \pi_\mu(a_t|s_t)}[Q^{\pi_\mu}(s_t, a_t) - \beta \log \pi_\mu(a_t|s_t)]$, which is satisfied in maximum entropy reinforcement learning setting (Haarnoja et al., 2018).

Consider the soft Bellman equation:

$$
\begin{aligned}
Q^{\pi_\mu}(s_t, a_t) &= r(s_t|a_t) + \beta \log \pi_{\text{ref}}(a_t|s_t) + V^{\pi_\mu}(s_{t+1}) \\
&\leq r(s_t|a_t) + \beta \log \pi_{\text{ref}}(a_t|s_t) + \mathbb{E}_{a_{t+1} \sim \pi_{\text{new}}}[Q^{\pi_\mu}(s_{t+1}, a_{t+1}) - \beta \log \pi_{\text{new}}(a_{t+1}|s_{t+1})] \\
&\cdots \\
&\leq Q^{\pi_{\text{new}}}(s_t, a_t)
\end{aligned}
\tag{19}
$$

where we iteratively expand $Q^{\pi_\mu}$ by applying the soft Bellman equation and the bound provided in Eq 18. Since $\pi_{\text{new}}$ is an improved policy over $\pi_\mu$ (i.e., $Q^{\pi_{\text{new}}(s_t, a_t)} \geq Q^{\pi_\mu}(s_t, a_t)$), and $\pi_\theta$ is trained to closely match $\pi_{\text{new}}$, it follows that $\pi_\theta$ will also be an improved policy over $\pi_\mu$ provided the approximation is accurate. Formally, we have:

$$
V^{\pi_\theta}(s) \approx V^{\pi_{\text{new}}}(s) \geq V^{\pi_\mu}(s)
\tag{20}
$$

Since we only incorporate the correct trajectory from the historical data, we can assert that policy $\pi_\mu$ will not perform worse than the old policy $\pi_{\text{old}}$. Therefore, we have the relationship:

$$
V^{\pi_\theta}(s) \approx V^{\pi_{\text{new}}}(s) \geq V^{\pi_\mu}(s) \geq V^{\pi_{\text{old}}}(s)
\tag{21}
$$

The policy improvement process is well-established. By iteratively performing policy evaluation and policy improvement, the policy $\pi_\theta$ can be made to converge toward the optimal policy $\pi^*$. If the data in the replay buffer is not utilized, the same theoretical process holds. However, in this case, $V^{\pi_\mu}$ and $Q^{\pi_\mu}$ are replaced by $V^{\pi_{\text{old}}}$ and $Q^{\pi_{\text{old}}}$.

We conduct an experiment to evaluate the effectiveness of using $V^{\pi_\mu}$ instead of $V^{\pi_{\text{old}}}$ in WEBRL training. In this experiment, the policy $\pi_\theta$ is trained using data from two sources: rollout trajectories during the current phase and past experiences stored in the replay buffer. For the critic, we compare two training settings: training with data from both rollout trajectories and experiences in the replay buffer versus using only rollout trajectories. The results, as presented in Figure 8, reveal that the model performs better when the critic is trained with data from both the replay buffer and the rollout trajectories. This observation underscores the significance of integrating experiences from prior successful trajectories into the critic's training process, which can lead to more effective policy improvement.

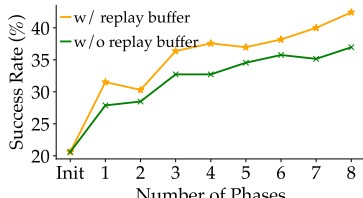

Figure 8: Comparison of using $V^{\pi_\mu}$ and $V^{\pi_{\text{old}}}$.

### A.3 COMPARISON

The comparison result between the policy update algorithm of WEBRL and other algorithms is shown in Table 4.

Table 4: Comparison between WEBRL and different algorithms based on various criteria.

| Algorithm | Off-policy | Pair-wise Data | KL-constrained Target | Reduce Error Prob |
|-----------|:----------:|:--------------:|:---------------------:|:-----------------:|
| WEBRL | ✔ | ✘ | ✔ | ✔ |
| DPO | ✔ | ✔ | ✔ | ✔ |
| PPO | ✘ | ✘ | ✔ | ✔ |
| AWR | ✔ | ✘ | ✘ | ✘ |

**Comparison with DPO (Rafailov et al., 2024b).** DPO also solves for the optimal value of eq. 1, obtaining the relationship between the optimal policy $\pi^*$ and the reward $r$. Rather than explicitly solving for the reward $r$, DPO employs contrastive learning by constructing paired data to build the learning target of $\pi_\theta$. This approach allows DPO to bypass the need to estimate $r$. In contrast, our approach explicitly fits the value function $V$ to guide the optimization of $\pi_\theta$, making it converge towards the optimal policy $\pi^*$. **However, DPO requires pair-wise data, which is challenging to**

**obtain in web tasks.** This is primarily because implementing state backtracking on web pages is difficult, making it hard to collect outcomes of different actions on the same page state.

**Comparison with PPO (Schulman et al., 2017).** Eq.1 is also the target of RLHF. Prior work often employs PPO to optimize this objective. **However, as an on-policy algorithm, PPO has low sampling efficiency and requires large amounts of data for stable improvement**. This limitation makes it unsuitable for environments such as WebArena and real-world websites, where interaction is expensive and inefficient. Instead, our approach operates in an off-policy manner.

**Comparison with AWR (Peng et al., 2019).** AWR is one of the widely used offline RL algorithms. The loss function of AWR is $\arg\max_\theta \mathbb{E}_\nu \left[ \log \pi_\theta(a|s, I) \cdot \exp(A(s, a, I)/\beta) \right]$. **Compared to our method, AWR has a different optimization goal.** Our approach explicitly minimizes the KL divergence between the reference policy $\pi_{\mathrm{ref}}$ and the current policy $\pi_\theta$, while AWR focuses on constraining the KL divergence between the current policy and the behavioral policy derived from training data $\nu$ (instead of the reference policy $\pi_{\mathrm{ref}}$). Additionally, **AWR does not directly reduce the probability of incorrect actions.** In web tasks, sampled data often include unsuccessful traces. Ignoring such data wastes valuable information, while using it risks increasing the likelihood of incorrect actions. In contrast, our method effectively reduces the probability of incorrect actions.

# B  TRAINING DETAILS

## B.1  DETAILS OF WEBRL

**The agent observation** consists of three components:

- User instruction: The instruction provided by the user.
- Action history: A record of the actions the agent has previously taken.
- Webpage HTML: HTML content of the current webpage.

We process the HTML, simplifying its structure and assigning distinct element IDs to all clickable elements. This facilitates the model's ability to identify and indicate which specific element requires manipulation.

**The agent actions** are mostly similar to those defined in WebArena-Lite (Liu et al., 2024c), with some additional actions.

- Click: Clicks an element with a specific ID.
- Hover: Hovers over an element with a specific ID.
- Type: Types a message into an input box with a specific ID.
- Search: Types a message into an input box with a specific ID and presses Enter to initiate a search.
- Press: Emulates a specific keyboard key combination.
- Scroll: Scrolls the page up or down.
- Select_dropdown_option: Selects an option from a dropdown menu with a specific ID.
- New_tab: Opens a new tab in the current browser.
- Tab_focus: Switches focus to a browser tab at a specified index.
- Close_tab: Closes the current tab.
- Goto: Navigates to a specific URL.
- Go_back: Returns to the previous page.
- Go_forward: Moves to the next page if available.
- Exit: Terminates the operation, returns the response, and exits.

To provide more detailed information about the action, we include comments labeled with "# Element:" in the action, which describe the operated element. Similarly, we include comments labeled "# Note:", which quote relevant information from the current webpage that supports completing the instruction. An example of the agent's specific input and output is shown in Figure 9. The input is the

observation, while the output specifies the action to be performed on the webpage. The "element" argument identifies the target element for the action. More examples can be found in Appendix F.

**The detailed training process of WEBRL** is shown in Algorithm 1. First, we perform supervised fine-tuning using the WebArena-Lite training dataset. We then initialize the replay buffer and failure set by running the SFT-trained model on instructions corresponding to the WebArena-Lite training dataset. Subsequently, in each phase of the self-evolving curriculum reinforcement learning process, 500 new instructions that meet the filtering criteria are selected from those generated by GPT-4o. Both newly generated interaction data on these instructions and historical data with perplexity between 1/0.95 and 1/0.5 from the replay buffer are used to train the actor and critic. The amount of historical data used is limited to twice the size of the interaction data. The hyperparameters employed in WEBRL are presented in Table 5.

## B.2 DETAILS OF BASELINES

For RL-based baselines (except DigiRL), the interaction data from WEBRL's first phase is used, while DigiRL is trained using the first-phase instructions in an online learning setup. Hence, except for DigiRL, the other RL baselines fall under offline reinforcement learning.

We reproduce the same framework used in DigiRL within the WebArena environment. Specifically, we use the same components, including the AWR method, the instruction-level and step-level value functions, and the replay buffer described in DigiRL. To apply DigiRL in WebArena-Lite, the main modifications we make are adjusting the data format to align with WEBRL and tweaking certain hyperparameters. DigiRL also conducts 8 rounds of interaction and training. The hyperparameters employed in those baselines are presented in Table 5.

---

Agent Input

Task Instruction: List orders where the status is Canceled and the recipient is Jennifer White, ordering them by decreasing total value.

Round 0
user:
List orders where the status is Canceled and the recipient is Jennifer White, ordering them by decreasing total value.
assistant:
# Element: the 'Sales' menu item on the left sidebar, second from the top
do(action="Click", element="2")

Round 1
user:
** Simplified html **
assistant:
# Element: the 'Orders' link
do(action="Click", element="4")

Round 2
user:
<html data-bbox="0,0,1280,720"><body data-bbox="0,0,1280,968"><div data-bbox="0,0,88,721"> <ul data-bbox="0,75,88,646">
 Dashboard 
<-- middle content omitted -->
<button id="24" type="button" data-bbox="548,305,18,17"> </button> </li> </ul>
*<button id="25" type="button" data-bbox="1167,304,83,19"> Clear all </button> </div>*
<-- middle content omitted -->
</div> </main> </div> </body> </html>
assistant:

Agent Output

# Element: the 'Clear all' button
do(action="Click", element="25")

Figure 9: The input and output format of WEBRL and baselines, where the input is composed of task instruction (in green), action history (in blue), and HTML of the current webpage (in orange). The output (in red) is the action taken on the current webpage.

---

## B.3 DETAILS OF ORM

WebArena-Lite provides 1,186 training samples each comprising an instruction, a trajectory, and a reward function. To train the $\mathcal{M}_{ORM}$, we first enhance the WebArena-Lite training dataset by rewriting the instructions. Then, we collect rollouts from all baseline methods on this augmented dataset. These rollouts, combined with the original WebArena-Lite training samples, form the com-

plete training dataset for the $\mathcal{M}_{\text{ORM}}$, with a total of 12,200 samples. We use the associated reward function to label these trajectories, identifying whether they successfully complete the task. These trajectories are subsequently used to train the ORM, with the specific hyperparameters listed in Table 6. The prompt for $\mathcal{M}_{\text{ORM}}$ is shown in Figure 15. $\mathcal{M}_{\text{ORM}}$ is required to produce either "YES" or "NO" as its output. To determine the evaluation result, we compare the probabilities assigned to "YES" and "NO" and select the one with the higher probability. We need to train $\mathcal{M}_{\text{ORM}}$ because new tasks that do not have predefined reward functions will be generated in the subsequent self-evolving curriculum reinforcement learning process. We rely on the trained $\mathcal{M}_{\text{ORM}}$ to judge whether each task is successfully completed.

---

**Algorithm 1** WEBRL

---

**Input:** WebArena-Lite training set $D_0$ and corresponding instruction set $I_0$, Base model $\mathcal{M}_{\text{base}}$, Replay buffer $\mathcal{B}$, Failure set $\mathcal{F}$, Reward model $\mathcal{M}_{\text{ORM}}$, Number of phases $N$, Number of instructions per phase $K$

   **Part 1. SFT Training**

  1: $\mathcal{M}_{\text{SFT}} \longleftarrow$ perform supervised fine-tuning on model $\mathcal{M}_{\text{base}}$ with dataset $D_0$
  2: $D_{\text{rollout}} \longleftarrow$ rollout trajectories of $\mathcal{M}_{\text{SFT}}$ on $I_0$
  3: $D_{\text{success}}, D_{\text{fail}} \longleftarrow$ evaluate $D_{\text{rollout}}$ using WebArena-Lite reward function
  4: $\mathcal{B} \longleftarrow D_0$ # Initialize replay buffer with successful trajectories
  5: $\mathcal{F} \longleftarrow D_{\text{fail}}$ # Initialize failure set with instructions of failing trajectories

   **Part 2. Self-evolving Curriculum RL**

  6: $\pi_1 \longleftarrow \mathcal{M}_{\text{SFT}}$ # Initialize actor/policy
  7: $V_1 \longleftarrow \mathcal{M}_{\text{SFT}}$ with a randomly initialized value head # Initialize critic
  8: **for** $n$ in $1...N$ **do**
  9:    $I_n \longleftarrow \{\}$
10:    **while** size($I_n$) $\leq K$ **do**
11:       $I_{\text{generation}} \longleftarrow$ instructions generated by GPT-4o with instructions from $\mathcal{F}$ as examples # instruction generation process
12:       $I_{\text{filter}} \longleftarrow \text{filter}(I_{\text{generation}})$ # instruction filtering process
13:       $I_n \longleftarrow I_n \cup I_{\text{filter}}$
14:    **end while**
15:    $D_{\text{rollout}} \longleftarrow$ rollout trajectories of $\pi_n$ on $I_n$
16:    $D_{\text{success}}, D_{\text{fail}} \longleftarrow$ evaluate $D_{\text{rollout}}$ with $\mathcal{M}_{\text{ORM}}$ # use $\mathcal{M}_{\text{ORM}}$ to label the rollout trajectories
17:    $D_{\text{experience}} \longleftarrow$ experiences from $\mathcal{B}$ with perplexity computed by $\pi_n$ between $\frac{1}{0.95}$ and $\frac{1}{0.5}$
18:    $\pi_{n+1}, V_{n+1} \longleftarrow \text{train}(\pi_n, V_n, D_{\text{rollout}} \cup D_{\text{experience}})$ # use loss functions from eq. 4 and eq. 7
19:    $\mathcal{B} \longleftarrow \mathcal{B} \cup D_{\text{success}}$
20:    $\mathcal{F} \longleftarrow \mathcal{F} \cup D_{\text{fail}}$
21: **end for**

---

## C  OTHER QUANTITATIVE EXPERIMENTS

Figure 10 illustrates the performance variation curves of Llama3.1-8B trained with WEBRL on each website. It can be seen that in all the sites except for Map, there is a clear upward trend. However, in the case of Map, there is an initial upward trend followed by a decline. We hypothesize that the final decline is caused by a significant increase in OSS and CMS implementation, which creates a trade-off. This trade-off leads to a performance drop in Map and a slight decline in GitLab.

Figure 11 illustrates the error bars for WEBRL across each phase. We conduct four sets of experiments, each utilizing a distinct random seed. The results from these four sets of experiments are presented in Figure 11. Despite some fluctuations in accuracy, WEBRL consistently demonstrates an overall upward trend. Furthermore, its accuracy consistently surpasses all baseline methods after completing the final phase of training.

Figure 12 presents the success rate of WEBRL and DigiRL, both with and without the application of our self-evolving curriculum learning. The results indicate that incorporating curriculum learning significantly enhances the performance of DigiRL. However, its performance still remains below that of WEBRL. Conversely, when WEBRL is implemented without curriculum learning, its performance experiences a notable decline, yet it still slightly surpasses that of DigiRL.

Table 5: The hyperparameters we employ in WEBRL and baselines.

| Method | Hyperparameter | Value |
|---|---|---|
| SFT | learning rate | 1e-5 |
| | lr scheduler type | cosine |
| | warmup ratio | 0.1 |
| | batch size | 128 |
| | training epoch | 1 |
| | cutoff length | 16384 |
| Filtered BC | learning rate | 1e-6 |
| | lr scheduler type | constant |
| | batch size | 128 |
| | training epoch | 1 |
| | cutoff length | 16384 |
| | filtering threshold | 70th percentile |
| AWR | actor learning rate | 1e-6 |
| | actor lr scheduler type | constant |
| | critic learning rate | 1e-6 |
| | critic lr scheduler type | constant |
| | batch size | 128 |
| | discount factor | 0.9 |
| | actor training epoch | 1 |
| | critic training epoch | 1 |
| | cutoff length | 16384 |
| DigiRL | actor learning rate | 1e-6 |
| | actor lr scheduler type | constant |
| | critic learning rate | 1e-6 |
| | critic lr scheduler type | constant |
| | instruction value function lr | 1e-6 |
| | instruction value function lr scheduler type | constant |
| | batch size | 128 |
| | discount factor | 0.9 |
| | actor training epoch | 1 |
| | critic training epoch | 1 |
| | instruction value function epoch | 1 |
| | rollout temperature | 1 |
| | replay buffer size | 10000 |
| | cutoff length | 16384 |
| WEBRL | actor learning rate | 1e-6 |
| | actor lr scheduler type | constant |
| | critic learning rate | 1e-6 |
| | critic lr scheduler type | constant |
| | batch size | 128 |
| | discount factor | 0.9 |
| | actor training epoch | 1 |
| | critic training epoch | 1 |
| | rollout temperature | 1 |
| | replay buffer size | 100000 |
| | cutoff length | 16384 |

Figure 13 illustrates the error bars for both the baselines, DigiRL w/ CL, and WEBRL. Notably, the performance of WEBRL and DigiRL with curriculum learning (CL) shows a higher standard deviation, which can be attributed to the inherent randomness in task generation. Despite this variability, WEBRL consistently outperforms all baseline methods, demonstrating its effectiveness.

Figure 14 compares two settings of the replay buffer: one containing only past successful trajectories and the other including all past trajectories, both successful and failed. The results show that including failed trajectories leads to slower and more volatile performance improvement. This issue arises because the advantage of an action in historical trajectories is estimated based on the trajectory's final reward. Including failed historical trajectories introduces a potential problem: even if

Table 6: The hyperparameters we employ to train the ORM.

| Hyperparameter | Value |
|---|---|
| learning rate | 5e-6 |
| lr scheduler type | cosine |
| warmup ratio | 0.1 |
| batch size | 128 |
| training epoch | 4 |
| cutoff length | 16384 |

certain actions in a failed trajectory are correct, they are likely to receive a negative advantage when the trajectory's final reward is zero. This reduces the likelihood of selecting these correct actions, thereby hindering effective policy improvement and leading to degraded performance.

We employ a variety of prompts to assess the stability of the curriculum learning strategy in WEBRL. Specifically, we utilize three distinct prompts: Prompt 1 (in Figure 16), Prompt 2 (in Figure 17), and Prompt 3 (in Figure 18). The training results for these prompts, measured after eight phases of training, are presented in Table 7. The results demonstrate that when using a variety of prompts to generate tasks, WEBRL consistently delivers strong performance, highlighting the stability of our method.

Table 7: Performance of different prompts for task generation.

| Prompt 1 | Prompt 2 | Prompt 3 |
|---|---|---|
| 42.4 | 40.6 | 43.6 |

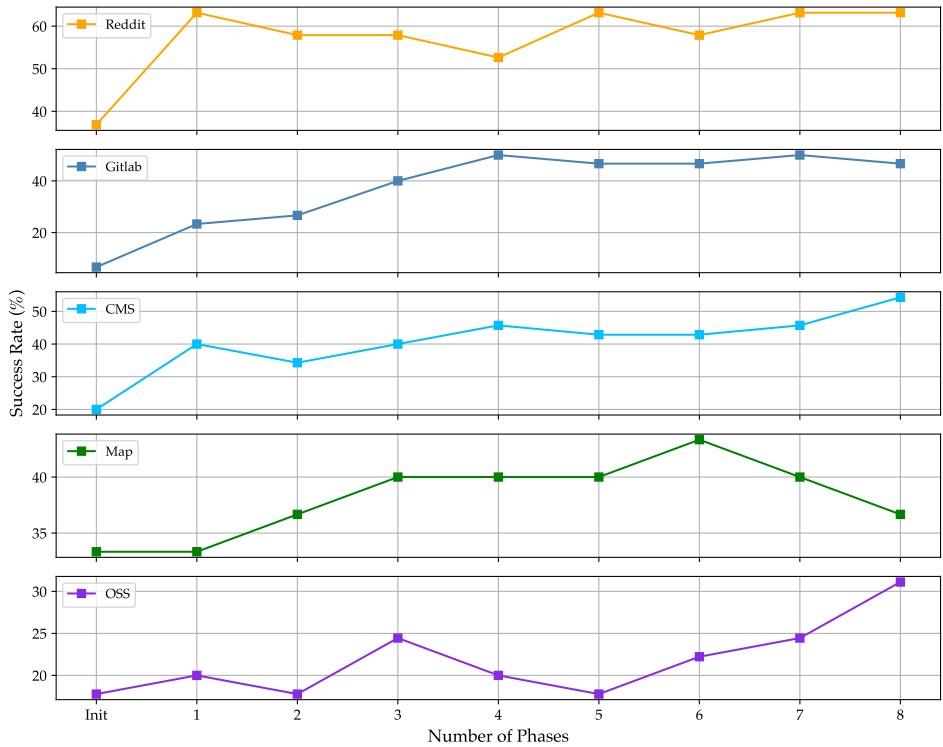

Figure 10: Performance variation curves of Llama3.1-8B on each website under WebRL training

# D  PROMPTS EMPLOYED IN WEBRL

The prompt used for $\mathcal{M}_{ORM}$ is shown in Figure 15. We require the model to output "YES" or "NO" to determine whether a certain trajectory successfully completes the corresponding task. Considering the limited context size, we only input the HTML content of the last state.

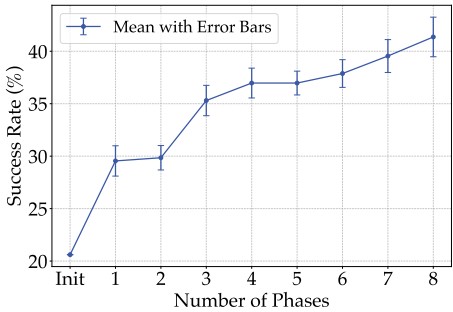

Figure 11: Error bars of WEBRL for each phase.

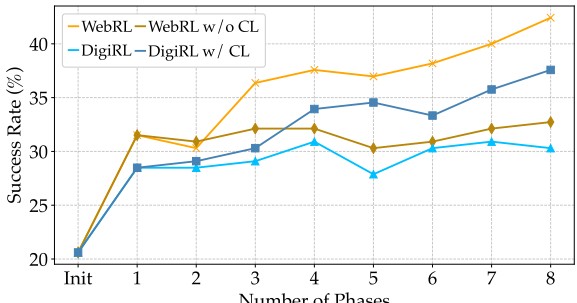

Figure 12: Comparison of WEBRL and DigiRL with and without curriculum learning (CL).

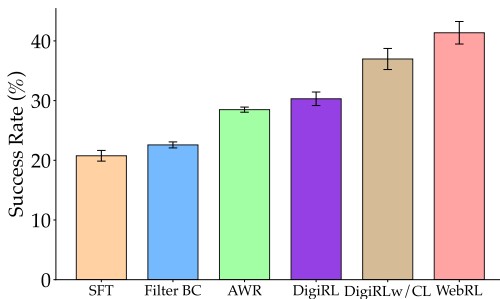

Figure 13: Error bars of baselines, DigiRL w/ CL, and WEBRL.

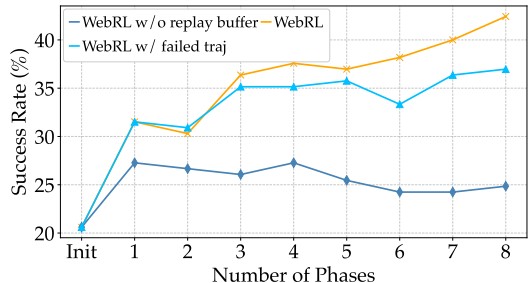

Figure 14: Comparison of WEBRL and WEBRL with failed trajectories in the replay buffer.

The prompt for generating new instructions is presented in Figure 16. We use tasks that the model fails to complete successfully in previous phases as seeds for generating new tasks of similar difficulty but with greater variety. The prompts shown in Figure 17 and Figure 18 represent alternative versions of the prompts used for task generation.

The simple prompt we use to test models including GPT-4-Turbo, GPT-4o, Llama3.1-8B-Instruct, and Llama3.1-70B-Instruct is shown in Figure 19. In this prompt, we define the feasible actions and provide illustrative examples. Additionally, we outline a set of requirements in the "REMEMBER" part to guide the model's behavior.

The prompt we use to filter the instructions generated by GPT-4o is presented in Figure 20. In this prompt, we provide a rule defining the feasibility of instructions across the five websites in WebArena.

## E  CASE STUDY

Figure 21 presents some instructions generated by the self-evolving curriculum learning strategy across different phases. Although these instructions are grouped by phase, the instructions shown in phase $i + 1$ are not necessarily generated using the instructions from phase $i$ as seeds. As the phase increases, two types of augmentation occur for previously incomplete instructions. In one case, new instructions with similar task requirements or lower difficulty levels are generated. These new instructions help the model to successfully complete previously unfinishable tasks by allowing it to practice and build competence with the easier or related tasks first. For example, in phase 2, the instruction improves upon the phase 1 instruction by offering a clearer task description and explicitly requiring results in "yearly interval". This enhancement enables the model to complete the task successfully by removing ambiguity. Additionally, changing the description of the instruction can improve agent exploration. With the positive feedback from the clarified phase 2 instruction, the model can better understand and accurately perform the original phase 1 task as well. In another case, tasks with increased complexity and diversity are generated. This task complication facilitates a continuous improvement in the model's capabilities by challenging its performance boundaries.

---

**System Prompt:**
You are an expert in evaluating the performance of a website navigation agent. The agent is designed to help a human user navigate the website to complete a task. Given the user's intent, the agent's action history, and the final state of the screen, your goal is to decide whether the agent's execution is successful or not. You must respond with YES or NO.

---

**User Prompt:**
The User Intent: {instruction}
Action History: {action history}
The Current Screenshot: {html of last state}

---

Figure 15: Prompts for $\mathcal{M}_{\text{ORM}}$ to assess the completion of Instructions.

---

**User Prompt:**
You are a smart task creator for a website intelligent assistant. Your goal is to generate clear and practical tasks that the assistant can assist people with when they use {web} in their daily lives. These tasks should encompass a wide range of possible instructions and questions that may arise when using {web} website.

Your need to draw inspiration from the #Given Task# to create new tasks. These new tasks should belong to the same domain as the #Given Task# but be more diverse. The difficulty level of the #Created Task# should be similar to that of the #Given Task#. The #Created Task# must be reasonable, understandable and realistic. '#Given Task#', '#Created Task#', 'given task' and 'created task' are not allowed to appear in #Created Task#.

You need to make sure, as much as possible, that the variable names in the #Created Task#, like the place name, username, and product name, are consistent with #Given Task#. You need to avoid making up some new variable names yourself.

#Given Task#
{task examples}

#Created Task#

---

Figure 16: Prompts for instruction generation.

Therefore, the process of instruction generation exhibits such a pattern: for tasks that the model is unable to perform, analogous tasks are created to provide incremental steps that facilitate learning how to accomplish that type of task. Furthermore, tasks that remain challenging for the model are also generated and undergo the aforementioned iterative process. Through this cycle of challenge and refinement, the model's capabilities gradually expand, enabling it to handle increasingly complex tasks over time.

**User Prompt:**
You are a smart task creator for a website intelligent assistant. Your goal is to generate clear and practical tasks that the assistant can assist people with when they use {web} in their daily lives. These tasks should encompass a wide range of possible instructions and questions that may arise when using {web} website.

Your need to draw inspiration from the #Given Task# to create new tasks. These new tasks should belong to the same domain as the #Given Task# but be more diverse. The difficulty level of the #Created Task# should be similar to that of the #Given Task#. The #Created Task# must be reasonable, understandable and realistic. '#Given Task#', '#Created Task#', 'given task' and 'created task' are not allowed to appear in #Created Task#.

**Guidelines:**
- Use a variety of phrasing styles to avoid repetitive expressions.
- Use variable values that same as those in the provided task examples, such as place names, usernames, subreddit name, and product names. Avoid inventing entirely new variable values.
- Maintain the same or similar difficulty level as the #Given Task#. Tasks can be slightly more or less challenging but should stay within a reasonable range.
- Before each generated task, first output the thought process.

#Given Task#
{task examples}

#Created Task#

Figure 17: Prompts for instruction generation (version 2).

**User Prompt:**
You are a highly intelligent task creator for a website assistant. Your role is to generate clear and practical instructions or questions that users might need help with when interacting with the `{web}` service in their daily lives. The tasks you create should reflect a wide range of possibilities and be realistic for users to request assistance with.

You will be provided with **Example Tasks** to draw inspiration from. Your job is to create new tasks within the same domain as the examples, ensuring diversity in phrasing, purpose, and scope.

### **Guidelines for Task Creation:**
1. **Extract Variable Values:**
  - Analyze the provided examples and extract the key variable values, such as names, places, usernames, subreddit names, product titles, or other specific terms.
  - Use these variable values when constructing new tasks. Avoid inventing entirely new values.

2. **Maintain Domain and Difficulty:**
  - Ensure the new tasks belong to the same domain as the example tasks.
  - Match the difficulty level of the example tasks, allowing for slight variation (e.g., slightly easier or harder) while keeping them reasonable and realistic.

3. **Ensure Clarity and Variety:**
  - Format each task clearly using backticks (`) for task descriptions.
  - Use varied phrasing to avoid repetitive expressions and make tasks more engaging and dynamic.

4. **Avoid Explicit Reference to Examples:**
  - Do not include phrases like "Example Tasks" or "New Tasks" in your output.
  - Tasks should stand independently, without explicit references to the examples or the task creation process.

### **Example Tasks**
{task_examples}

### **Steps to Follow:**
1. **Extract Variable Values:**
  - List the key variable values from the example tasks.
2. **Generate New Tasks:**
  - Using the extracted variable values, construct a set of new tasks within the same domain.

**Output Format:**
1. Extracted Variable Values:
  ```
  - Variable 1: [value]
  - Variable 2: [value]
  - ...
  ```

2. New Tasks:
  ```
  `Task 1 description`
  `Task 2 description`
  ...
  ```

Figure 18: Prompts for instruction generation (version 3).

**System Prompt:**
# Setup
You are a professional web browsing agent assistant that can fulfill user's high-level instructions. Given Simplified html of the browsed webpage at each step, you plan operations in python-style pseudo code using provided functions, or customize functions (if necessary) and then provide their implementations.
# More details about the code
Your code should be readable, simple, and only **ONE-LINE-OF-CODE** at a time, avoid using loop statement and only use if-else control if necessary. Predefined functions are as follow:

```
def do(action, argument, element):
    """A single browsing operation on the webpage.
    Args:
        :param action: one of the actions from ["Click", "Right Click", "Type", "Search", "Hover", "Scroll Up",
        "Scroll Down", "Press Enter", "Switch Tab", "Select Dropdown Option", "Wait"].
        :param argument: optional. Only for "Type", "Search", "Switch Page", and "Select Dropdown Option",
        indicating the content to type in, page number(start from 0) to switch, or key to press. "Search" action is
        equivalent to "Type" action plus "Enter" key press.
        :param element: optional. Only for "Click", "Right Click", "Type", "Search", "Select Dropdown Option",
        and "Hover". Should be specific element id in the html.
    Returns:
        None. The webpage will be updated after executing the action.
    """

def exit(message):
    """Ending the browsing process if the assistant think it has fulfilled the goal.
    Args:
        :param message: optional. If user's instruction is a question, return assistant's answer in the message
        based on the browsing content.
    Returns:
        None.
    """

def go_backward():
    """Go back to the previous page.
    """

def go_forward():
    """Go forward to the next page.
    """
```

Here are some examples:
- # Element: the 'REPORTS' section on the left sidebar
do(action="Click", element="7")
- # Element: the 'Period' dropdown, middle center
do(action="Select Dropdown Option", argument="Month", element="20")
[part of all examples in the used prompt]

REMEMBER:
- only **ONE-LINE-OF-CODE** at a time
- Don't generate an operation element that you do not see in the screenshot.
- Use "# Element" to describe the element you choose in the html.
- Use '# Note' to record information useful to answer the instruction if needed.
- If you find yourself fallen into some sort of loop, try to use another method or change your action.
- If you think a page is still loading or still playing animation and you want to wait a while, use "Wait" action.
- You are acting in a real world, try your best not to reject user's demand. Solve all the problem you encounter.
- If you think you didn't get expected webpage, you should try using more precise and locative description of the element.
- You should **NEVER** try to use the browser's address bar at the top of the page to navigate.
- Your answer shouldn't be in a code snippet format. Just write the function name and its arguments.
- If you use do function to perform "Click", "Right Click", "Type", "Search", "Select Dropdown Option", and "Hover", the param element must not be None.
"'

**User Prompt:**
Task Instruction: {instruction}
Action History: {action history}
The Current HTML: {html of last state}

Figure 19: The simple prompt employed in baselines.

**User Prompt:**
You are a task filtering expert, and you need to determine whether a given task is feasible or not.
These tasks are primarily distributed across the following five platforms: MAP (OpenStreetMap), Reddit, GitLab, CMS (online store content management system), and OSS (OneStopShop). You need to make judgments based on the following criteria:

1. For tasks in MAP:
Tasks that are supported by MAP itself are feasible unless they involve the following goals, which are deemed infeasible:
- Viewing traffic flow, accidents, and road closure information.
- Checking the weather conditions of a specific location.
- Marking and saving favorite locations, such as home, work, or travel destinations.
- Sharing real-time location.
- Making reservations and bookings, such as restaurant reservations or hotel bookings.
- Flight and train inquiries.
- Viewing event locations and activities, such as concerts, exhibitions, and their details.

2. For tasks in Reddit:
Tasks supported by Reddit is feasible.

3. For tasks in GitLab:
Tasks supported by GitLab is feasible.

4. For tasks in CMS:
Tasks that are supported by CMS itself are feasible unless they involve the following goals, which are deemed infeasible:
- Sending order information to customers via email.
- Automatically generating e-invoices for customers.
- Handling customer returns or refund requests.
- Supporting profile updates.
- Recommending products based on customer behavior.

5. For tasks in OSS:
Tasks that are supported by OSS itself are feasible unless they involve the following goals, which are deemed infeasible:
- Filtering out discounted products.
- Modifying the delivery address for a product.
- Adding payment information such as credit cards, e-wallets, or bank transfers.
- Displaying order status (e.g., pending payment, to be shipped, in transit, completed).
- Providing order tracking functionality to view real-time logistics.
- Offering online customer service to answer questions.
- Supporting after-sales service requests, such as refunds or repairs.

You should evaluate the following tasks based on these rules and respond in the following format:
Reason + Answer (where the answer is "[YES]" or "[NO]").

The website of the task is: {web}
The task is:{task}

Figure 20: The prompt used to filter instructions.

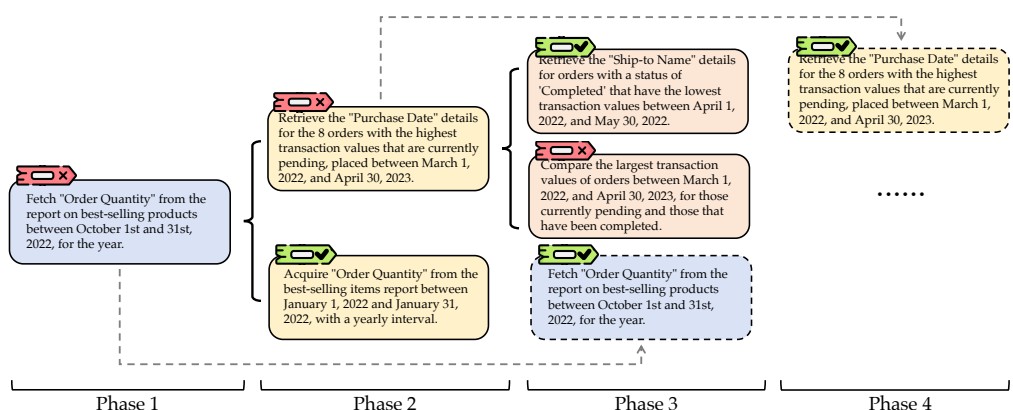

Figure 21: Examples of instructions generated in different phases under self-evolving curriculum learning.

# F    QUALITATIVE EXAMPLES

We list one example of WEBRL on each of the five sites in WebArena.

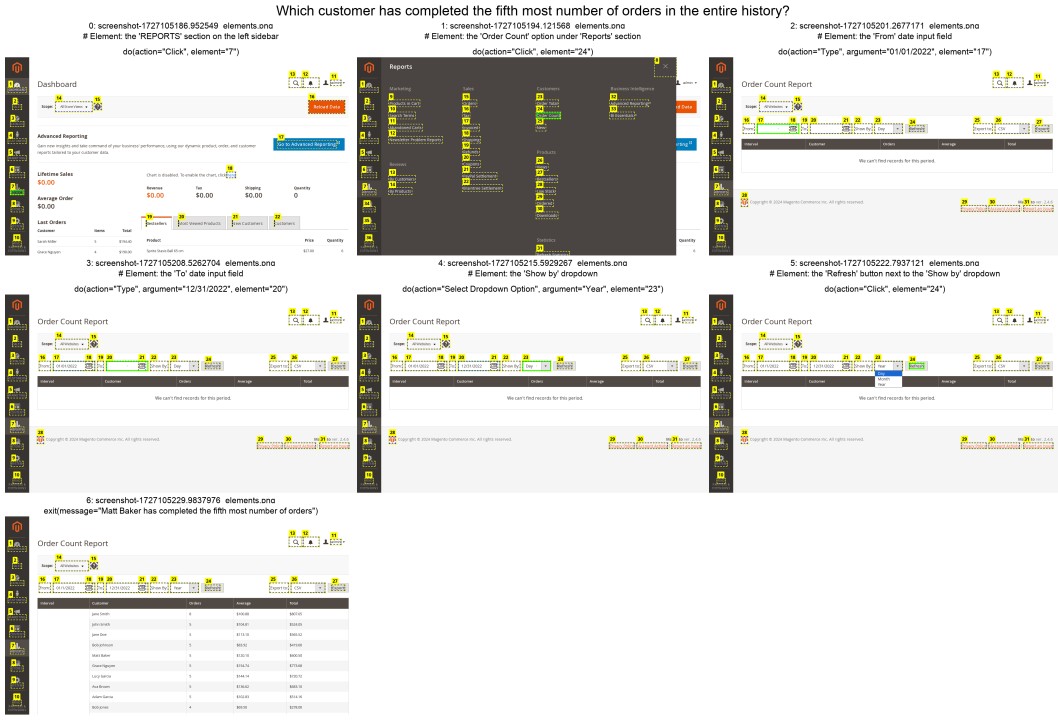

Figure 22: CMS Example.

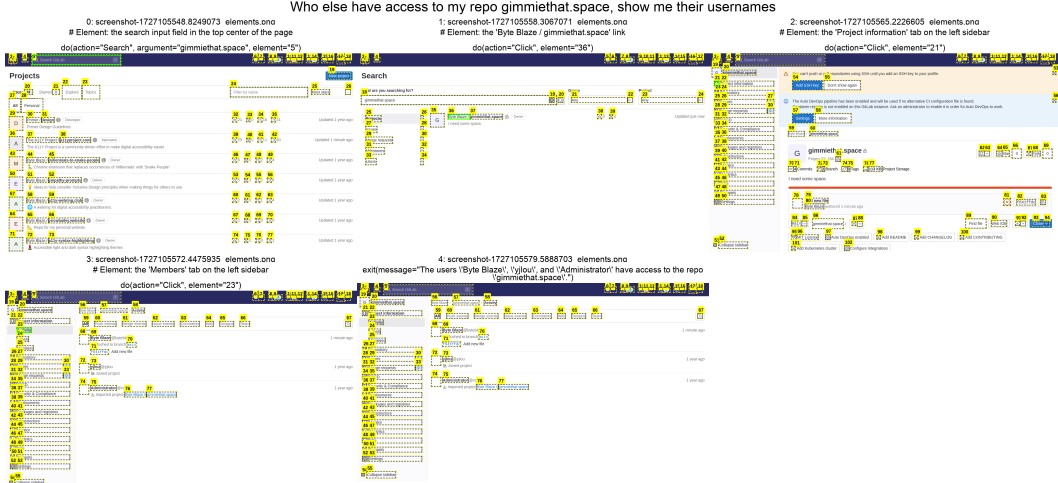

Figure 23: Gitlab Example.

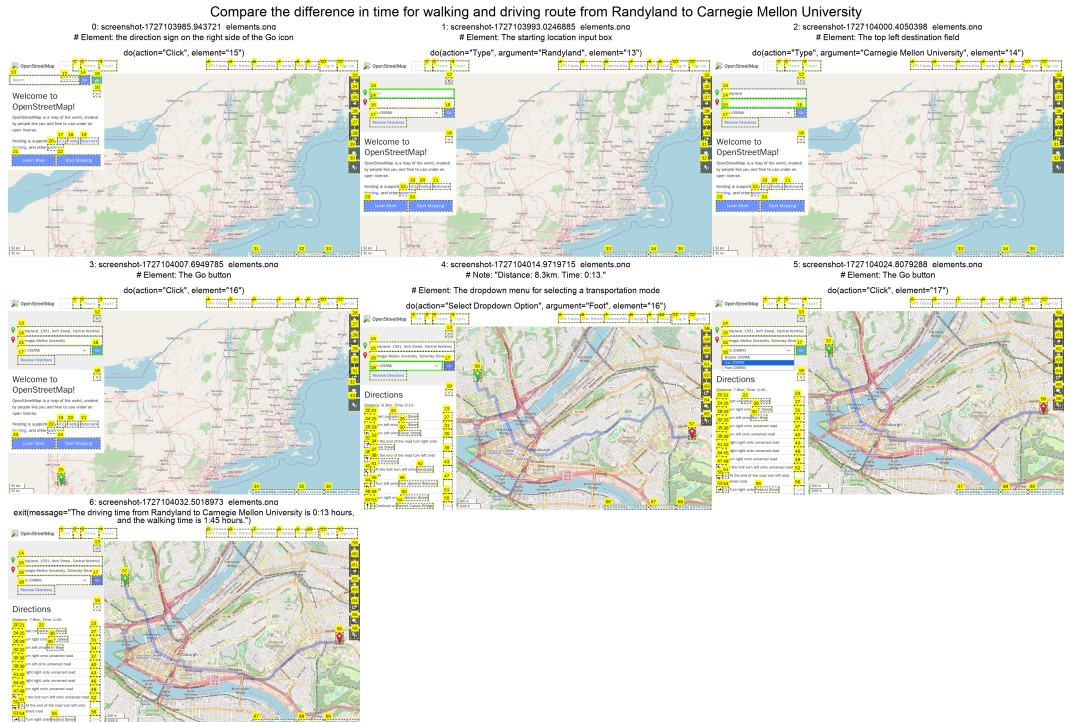

Figure 24: MAP Example.

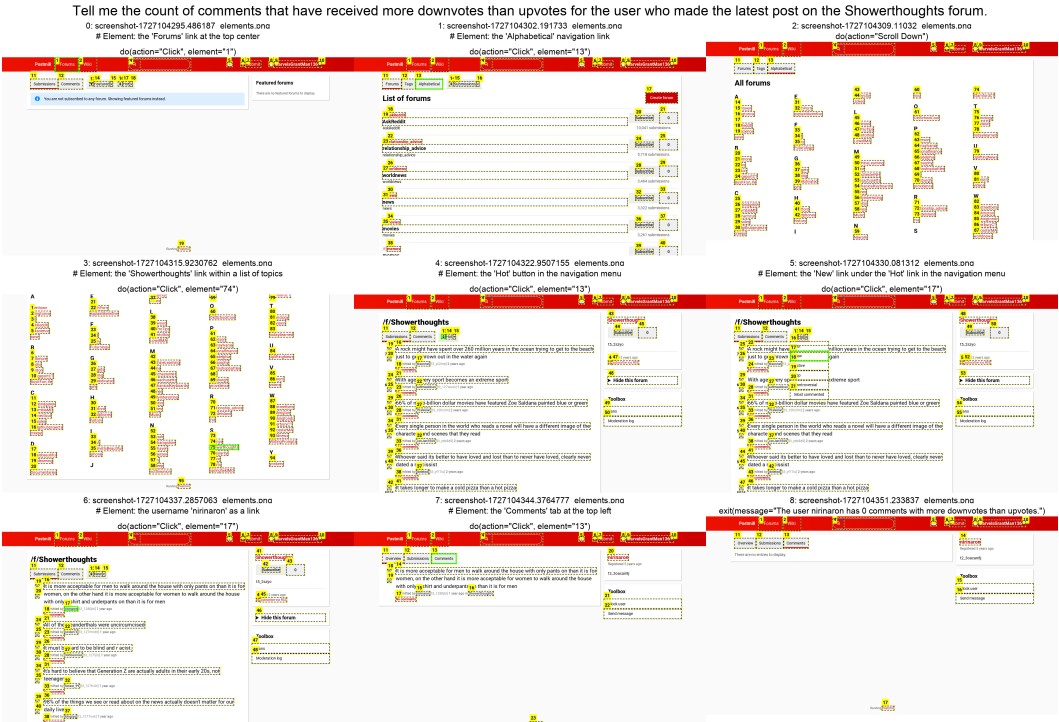

Figure 25: Reddit Example.

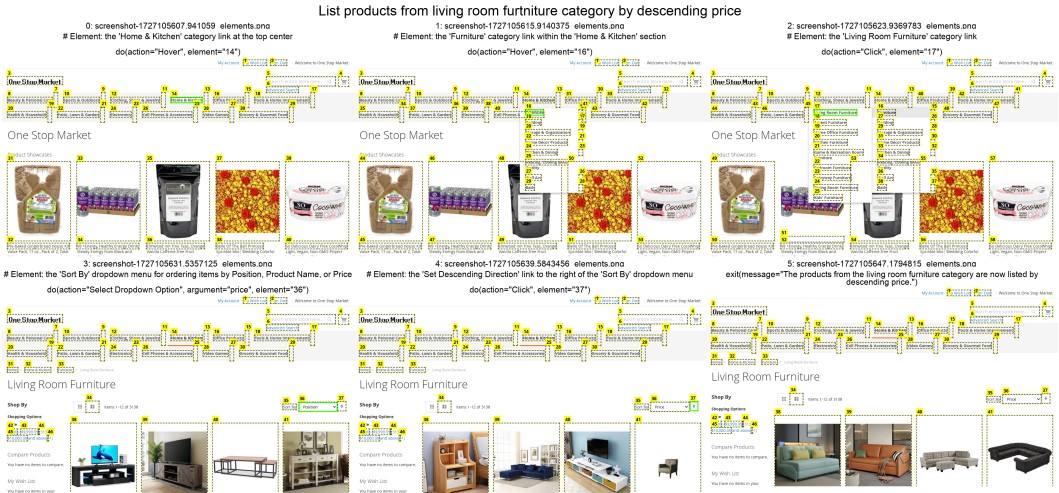

Figure 26: OSS Example.

