# OpenReview forum: "WebRL: Training LLM Web Agents via Self-Evolving Online Curriculum Reinforcement Learning"
_ICLR.cc/2025/Conference — ICLR 2025 Poster_

### Official Review · Reviewer_gjpa · 2024-10-29

**Soundness:** 2
**Presentation:** 1
**Contribution:** 2
**Rating:** 6
**Confidence:** 4

**Summary:**

This paper proposed WebRL, an online RL framework to train LLM-based web agents. It applies a curriculum learning for agents’ continual improvement with a trained outcome reward model. It applies a KL divergence to prevent the actor from deviating too much from the previous policy and a replay buffer to prevent catastrophic forgetting. Empirical studies show that WebRL outperformed baseline methods on WebArena-Lite benchmark and also has the ability to scale up.

**Strengths:**

1. WebRL is an online RL framework, which greatly saves the need for dataset preparation in SFT.
2. WebRL outperformed baseline methods on WebArena-Lite benchmark and also has the ability to scale up.

**Weaknesses:**

1. The authors claimed that WebRL is a self-evolving framework, mainly due to its curriculum learning module. However, the generated instructions need to be checked manually to eliminate unreasonable or impossible instructions. This manual check greatly weakens the term, self-evolving, which is one of the key contributions in this paper. It is also unknown to what extent this manual check impacts the overall training process.

2. The experimental results are not conducted fairly. All the RL-based methods, except for WebRL are only trained on the generated 500 instructions in the first phase, whereas WebRL is trained iteratively on multiple phases. Training curves (steps VS success rate) and error bars are also not provided in the paper. The RL-based methods need to report the results with several seeds.

3. The ablation study is not sufficient. In lines 342-346, the authors contribute the limitation of DigiRL to the lack of self-evolving curriculum learning, however, in the ablation study, the authors did not ablate the curriculum learning.

4. Limited training details are reported in the paper and appendix, such as the hyperparameter for the experiments, observation and action space, how to train the ORM, how to use the replay buffer, etc. I am highly concerned with the reproducibility of this work.

5. The writing needs improving, which is confusing and difficult to understand, especially in the introduction of ORM and curriculum learning, where curriculum learning should be introduced first in the second section. Due to the lack of examples, it is difficult to understand the difference of the generated instruction and the original tasks in the benchmark and the necessity of the ORM model.

6. A number of RL for LLMs works are not mentioned in the related works, which are not limited to GLAM[1], TWOSOME[2], RL4VLM[3] and ArCHer[4].

[1] Carta, Thomas, et al. "Grounding large language models in interactive environments with online reinforcement learning." International Conference on Machine Learning. PMLR, 2023.

[2] Tan, Weihao, et al. "True Knowledge Comes from Practice: Aligning Large Language Models with Embodied Environments via Reinforcement Learning." The Twelfth International Conference on Learning Representations.

[3] Zhai, Yuexiang, et al. "Fine-Tuning Large Vision-Language Models as Decision-Making Agents via Reinforcement Learning." arXiv preprint arXiv:2405.10292 (2024).

[4] Zhou, Yifei, et al. "Archer: Training language model agents via hierarchical multi-turn rl." arXiv preprint arXiv:2402.19446 (2024).

**Questions:**

1. What is the performance of WebRL without curriculum learning and DigiRL with curriculum learning?
2. In lines 77-79 “those in WebArena are typically of long horizons, with oracle solutions averaging 10 steps. This characteristic introduces substantial sparsity in the available signals during online exploration”. How can 10-step-long tasks be counted as long horizons in decision-making tasks?
3. What is the clear definition of phase?
4. Could you provide more analysis and insight into how RL mitigates the issue in lines 365-366?
5. Why discount factor is not applied in the value function?
6. In Lines 243-244, how to calculate the perplexity of the action?
7. In lines 304-305, where do the data come from?
8. What is the advantage of the loss shown in equation (6) compared to the standard loss function with KL?
9. How is the generalizbility of WebRL. Can the trained agent be deployed to other tasks or even to the real PC?

---

> ### Author Response · Authors · 2024-11-18
> **Response to Reviewer gjpa (Part 1 of 3)**
>
> Dear Reviewer gjpa,
>
> Thank you for your valuable feedback. We sincerely appreciate your insights and are fully committed to addressing these points to improve the quality of our work.
>
> ## Response to Weakness1: Reliance on human-filtering for curriculum steps
>
> Thank you for bringing this to our attention. We sincerely apologize for the confusion our previous expression may have caused. Here’s a clearer explanation of the manual check process:
>
> Initially, we conduct a manual review of the generated instructions and identify instructions that could not be completed in WebArena. Based on these observations, we develop a set of rules to filter out such instructions. These rules are then integrated into the prompt used by GPT-4o, enabling it to automatically exclude instructions that are not feasible in WebArena. Then, throughout the curriculum learning process, we use GPT-4o with this prompt to filter out infeasible instructions.
>
> For further clarity, we add this filter prompt to Figure 15 (Appendix D).  In a test of 100 instructions, GPT-4o with this prompt demonstrates 100% recall and 99% precision in identifying tasks deemed infeasible within WebArena.
>
> Additionally, this check process is essential because WebArena is a simulation environment with limitations compared to real websites. For instance, on OSS website, real shopping platforms allow modifications to delivery addresses, while WebArena does not. When using GPT to generate new tasks, it may inadvertently create tasks feasible on real websites but impossible in WebArena. Using unfeasible instructions as seeds for generating new ones could lead to an increasing proportion of unachievable tasks.
>
> ## Response to Weakness2: Deficiencies of the experiment
>
> **1\. Unequal number of training instructions**
>
> The comparison between WebRL and other baselines does not use an equal number of instructions. However, even when considering only a single training phase, WebRL achieves an accuracy of 31.5, outperforming these baselines. Additionally, as highlighted in the Introduction, the scarcity of high-quality training tasks poses a challenge for the development of WebAgents. To address this, we propose a self-evolving curriculum learning approach that generates tasks tailored to the agent's abilities. Since the baseline methods lack such a process, a direct comparison using the same instructions may not be appropriate.
>
> **2\. Lack of training curves and error bars**
>
> Thank you for your suggestion. We run WebRL four times and add error bars to Figure 10. While there are some fluctuations in accuracy, the results consistently exhibit an overall upward trend.
>
> The evaluation process for WebArena is highly time-consuming, requiring approximately 1 hour to complete a single evaluation round consisting of 165 tasks. This makes it challenging to plot training steps against success rates efficiently. Therefore, we present the success rate after each training phase in Figure 3 and show success rate variations for each website in Figure 9.
>
> ## Response to Weakness3: The ablation study is not sufficient
>
> Thanks for your suggestion. We add the result of WebRL w/o curriculum learning in Figure 3. The results indicate that when comparing WebRL to WebRL w/o CL, both methods demonstrate an overall upward trend due to the benefits of online learning. However, WebRL w/o CL shows slower progress and achieves a lower performance ceiling.
>
> ## Response to Weakness4: Limited training details
>
> We sincerely apologize for the confusion caused by the lack of training details. We add the following training details in our new updated manuscript:
>
> - Hyperparameters for WebRL and baselines in Table 5 and Table 6
> - Details of the observation and action space in Appendix B.1
> - The training process for the ORM in Appendix B.2
> - Pseudocode of WebRL in Algorithm 1 (lines 972-998), explaining how modules in WebRL are organized.

---

> > ### Author Response · Authors · 2024-11-18
> > **Response to Reviewer gjpa (Part 2 of 3)**
> >
> > ## Response to Weakness5: Writing Problems
> >
> > We sincerely apologize for any confusion caused by our manuscript. In response to your question, we would like to provide the following clarifications.
> >
> > **1\. ORM**
> >
> > In the field of large language models (LLMs), the reward model is used to evaluate the quality of the model's responses. These reward model typically includes the outcome-supervised reward model (ORM) and the process-supervised reward model (PRM). ORM evaluates task performance based on the final outcome, while PRM assesses the execution process, focusing on the quality of each step in completing the task.
> >
> > In our work, ORM is utilized to assess whether the agent’s rollout trajectory accomplishes a given task. ORM will output "YES" or "NO" to indicate the evaluation result, which is used as the binary reward signal (0 for failure and 1 for success). ORM is required because, during the training process of WebRL, the agent interacts with the environment on tasks that are newly generated by GPT.  Since WebArena cannot inherently provide rewards for these new tasks, ORM plays a critical role in implementing the scoring function, enabling effective evaluation.
> >
> > **2\. Curriculum learning**
> >
> > The self-evolving curriculum learning strategy generates new training instructions at each phase to guide the exploration and learning of the agent. Thank you for your suggestion. We move the introduction of curriculum learning to the first part of Section 2 in our updated manuscript.
> >
> >
> > **3\. Cases of the generated instruction**
> >
> > We provide examples of newly generated instructions during the self-evolving curriculum learning process in Appendix E. These instructions align with the domain of those in WebArena. However, the generated instructions are more diverse. This diversification of instruction enhances the model's exploration capabilities. Furthermore, the generated instructions exhibit varying levels of complexity. Instructions with greater complexity allow the model to continuously improve its capabilities. Simultaneously, instructions of lower complexity help the model to achieve positive rewards, facilitating progress toward completing more complex instructions effectively.
> >
> > ## Response to Weakness6: Lack of some related works
> >
> > Thank you for your suggestion. We appreciate your feedback. Regarding RL4VLM and ArCHer, we have already included these works in the related work section of our initial manuscript. In the updated manuscript, we incorporate GLAM and TWOSOME into the related work section as well.
> >
> > ## Response to Question1: Performance of WebRL w/o CL and DigiRL w/ CL
> >
> > We add comparison results for WebRL without curriculum learning and DigiRL with curriculum learning in the newly submitted manuscript (Figure 11). The results indicate that integrating curriculum learning significantly improves the performance of DigiRL. However, its performance still falls short of that achieved by WebRL. On the other hand, when WebRL is implemented without curriculum learning, its performance shows a decline but still outperforms DigiRL.
> >
> > ## Response to Question2: Long horizons
> >
> > Apologies for the ambiguity in our previous expression. To clarify, compared to existing device control platforms such as MiniWoB++ [1] (average of 3.6 steps), MIND2WEB [2] (average of 7.3 steps), and AITW [3] (average of 6.5 steps), WebArenta requires an average of 10 steps, making it a relatively long-horizon process.
> >
> > ## Response to Question3: Definition of phase
> >
> > The term "phase" refers to a distinct stage or step in the process of curriculum learning. Curriculum learning is often structured into phases to ensure an effective sequence of learning. In WebRL, each task generation marks the beginning of a new phase, as it represents a stage in the curriculum learning process where a new batch of tasks tailored to the agent's current skill level is generated. We use the term "phase" in reference to [4].
> >
> > ## Response to Question4: More analysis and insight
> >
> > - **Loss function**: RL introduces a weight to the loss of each action, while SFT treats all actions equally. For actions that occur frequently and consistently in the training data, such as "Scroll Down," SFT is prone to overfitting, which can lead to the model failing to determine when to stop and subsequently "getting stuck midway". RL, on the other hand, can mitigate this over-optimization by dynamically adjusting the loss weight for specific actions.
> >
> > - **Interaction**: RL also benefits from the ability to correct its own errors through interaction. If repeatedly executing an action results in an error, the model can learn to reduce the probability of taking that action again. In contrast, SFT lacks the capacity to adapt or self-correct during interaction.

---

> > > ### Author Response · Authors · 2024-11-18
> > > **Response to Reviewer gjpa (Part 3 of 3)**
> > >
> > > ## Response to Question5: Discount factor
> > >
> > > - **Critic**: We use the same critic loss function in DigiRL, which is well-suited for scenarios in web tasks where only the final step is rewarded. Unlike traditional TD error loss functions, this approach employs a cross-entropy loss function, without the need for the discount factor.
> > > - **Advantage**: We actually use the discount factor to estimate the advantage in our implementation with a value set at 0.9. However, for simplicity, we omit the discount factor during the derivation of the loss function. This omission does not impact the derivation process. For completeness, we provide the corresponding formula incorporating the discount factor in Appendix A.1.
> > >
> > > ## Response to Question6: Perplexity
> > >
> > > Perplexity is a metric in natural language processing (NLP) that measures the uncertainty of a language model for a piece of text. Lower perplexity indicates the model is better at predicting the text.
> > > The equation to calculate perplexity is:
> > >
> > > $\exp(-\frac{1}{n} \sum_{t=1}^n \log P(x_t|x_1, \cdots,x_{t-1}))$
> > >
> > > where $x_t$ represents a token in the sentence $(x_1, x_2, \cdots, x_n)$. The term $\sum _ {t=1}^n \log P(x _ t|x _ 1, \cdots,x _ {t - 1})$ sums up the log probabilities of each token in the sequence, reflecting the likelihood of generating the entire sentence.
> > >
> > > For the LLM agent, its action is a piece of text, $a = (x_1, x_2, \cdots, x_n)$. The probability of generating the action $P(a)$ is :
> > >
> > > $P(a) = \prod\_{t=1}^n P(x_t|x_1, \cdots, x_{t-1})$
> > >
> > > Given this, the perplexity of action $a$ can also be expressed as:
> > >
> > > $\text{PPL} = \frac{1}{\sqrt[n]{P(a)}}$
> > >
> > > ## Response to Question7: Source of Training data
> > >
> > > The training date is sourced from VisualAgentBench [5]. We email the author of VisualAgentBench to request the dataset.
> > >
> > > ## Response to Question8:  Compared to the standard loss function with KL
> > >
> > > Sorry, I'm not sure which standard loss function with KL you're referring to. Within the scope of my knowledge, the standard loss function with KL refers to the optimization target in RLHF [6]. Current research primarily utilizes PPO and DPO to address optimization objectives with KL constraints. The distinction between our approach and these methods is detailed below.
> > >
> > > - **Comparison with PPO**: PPO is an on-policy algorithm, whereas our approach is off-policy. In web scenarios, including WebArena or real-world web pages, interaction is expensive and inefficient, making on-policy algorithms unsuitable.
> > > - **Comparison with DPO**: While both are off-policy, DPO requires constructing pairwise data. In WebAgent scenarios, creating such data is challenging because implementing state backtracking on web pages is difficult, making it hard to collect outcomes of different actions on the same page state.
> > >
> > > If my understanding is incorrect, please feel free to clarify, and I'll be happy to address your question again.
> > >
> > > ## Response to Question9: Generalizability of WebRL
> > >
> > > WebRL demonstrates certain generalization capabilities. It can successfully handle straightforward web tasks, such as searching for specific information on Google or finding a location on Google Maps, even if those tasks are not included in its training data. However, it encounters challenges when performing more complex tasks, such as identifying and liking a particular tweet from a specific user.
> > >
> > > ----
> > >
> > > We appreciate the reviewer's suggestion and believe this additional analysis strengthens the validity of our work.
> > >
> > > If you have any further questions or concerns about our paper, please feel free to let us know. If our response has addressed your concerns, we would greatly appreciate it if you could kindly consider the possibility of revising our score. Thank you for your time and understanding.
> > >
> > > Best wishes,
> > >
> > > All authors
> > >
> > > ---
> > >
> > > **References**
> > >
> > > [1] Reinforcement learning on web interfaces using workflow-guided exploration. ICLR 2018
> > >
> > > [2] Mind2web: Towards a generalist agent for the web. NeurIPS 2023
> > >
> > > [3] Androidinthewild: A large-scale dataset for android device control. NeurIPS 2024
> > >
> > > [4] Curriculum Learning for Reinforcement Learning Domains: A Framework and Survey. JMLR 2020
> > >
> > > [5] Visualagentbench: Towards large multimodal models as visual foundation agents. ArXiv 2024
> > >
> > > [6] Training language models to follow instructions with human feedback. NeurIPS 2022

---

> > > > ### Comment · Reviewer_gjpa · 2024-11-20
> > > >
> > > > Thanks for the clarification and revision, which greatly improves the quality of the paper. I have raised my score but I still have a few concerns remained.
> > > >
> > > > 1. I agree with other reviewers that the algorithm does not seem to be an off-policy method. The advantage is calculated by equation 8, which is apparently in an on-policy way. Then the use of a replay buffer may be improper.
> > > >
> > > > 2. I still have great concerns with the reproducibility of the paper.
> > > >
> > > > - Both the curriculum learning and RL exploration introduce great randomness. The authors only provide the results of WebRL trained with 4 seeds, while all the other baselines and ablation studies are only run once. These results are not convincing enough, especially in the key ablation study where DigiRL also adopts curriculum learning. It is possible that WebRL happens to get a batch of good tasks and DigiRL happens to get a batch of bad tasks.
> > > >
> > > > - The provided code is not complete, so the results cannot be reproduced. Only offline training is available. The code for online interaction with the environment is missing. The code for curriculum learning is also independent of the training code. It seems that the authors manually start the curriculum learning and training process at every phase.

---

> > > > > ### Author Response · Authors · 2024-11-24
> > > > > **Response to Reviewer gjpa (Part 1 of 2)**
> > > > >
> > > > > Dear Reviewer 8quU,
> > > > >
> > > > > Thank you for your valuable suggestions. We sincerely appreciate your insights. We arrange our responses in the order your questions are raised, and hope they clarify your concerns.
> > > > >
> > > > > ## Response to Question1: Concern about the algorithm
> > > > >
> > > > > We provide a detailed proof in Appendix A.1 and A.2 demonstrating that our algorithm operates in the off-policy setting. For additional clarification, we expand upon this further below.
> > > > >
> > > > > **1\. Our algorithm is an off-policy algorithm**
> > > > >
> > > > > First, I would like to establish that our algorithm operates in the off-policy setting. Briefly, we first derive the optimal solution $\pi^*$ for the objective function (Eq. 1). Then we build the optimization target of $\pi_\theta$:
> > > > > $\arg\min\_{\theta} \mathbb{E}\_{\nu}[( \log\pi_\theta(a|s, I) - \log\pi^*(a|s, I))^2]$.
> > > > >
> > > > > This optimization target leads to the loss function:
> > > > >
> > > > > $\mathcal{L}(\pi_\theta) = \mathbb{E}\_{\nu} [( \beta \log \frac{\pi_\theta(a|s, I)}{\pi_{\text{ref}}(a|s, I)} - A^*(s, a, I))^2]$
> > > > >
> > > > > For any given state-action pair in $\nu$, $\pi_\theta$ is expected to match the target policy $\pi^*$. **There is no restriction on the data distribution $\nu$, indicating the algorithm can function in an off-policy setting.**
> > > > > Therefore, regardless of the method used to estimate the advantage, the strategy training process is off-policy. Even if the advantage estimation is on-policy, the overall algorithm remains off-policy.
> > > > >
> > > > > Kindly refer to Appendix A.1 for a detailed derivation.
> > > > >
> > > > > **2\. Using replay buffer for training critic and advantage is reasonable**
> > > > >
> > > > > Since we do not have access to $A*$, we approximate it using an advantage estimator $A$, which is derived from a value network $V$. Below we establish the feasibility of this approximation. Specifically, **we demonstrate that using
> > > > > $A$ to train the policy can lead to policy improvement.**
> > > > >
> > > > >
> > > > > The current policy is denoted by $\pi_\text{old}$. The training dataset $D$ is composed of trajectories sampled using $\pi_\text{old}$ as well as historical experiences stored in a replay buffer. The behavioral policy associated with the dataset $D$ is referred to as $\pi_{\mu}$. The corresponding value function and action-value function under $\pi_{\mu}$ are denoted as $V^{\pi_{\mu}}$ and $Q^{\pi_{\mu}}$,  respectively.
> > > > >
> > > > > **We demonstrate that minimizing the objective $\arg\min_\theta \mathbb{E}\_D[(\log_{\pi_\theta}(a_t|s_t) - \exp (\frac{Q^{\pi_\mu}(s_t, a_t) - V^{\pi_\mu}(s_t)}{\beta}))^2]$ can lead to policy improvement.**
> > > > >
> > > > > The new policy $\pi_\text{new}(a_t|s_t)$ is defined as:
> > > > > $\pi_\text{new}(a_t|s_t) = \exp\left(\frac{Q^{\pi_\mu}(s_t, a_t) - V^{\pi_\mu}(s_t)}{\beta}\right)$
> > > > >
> > > > > It can be shown that:  $D_\text{KL}(\pi_\text{new}||\pi_\text{new})
> > > > > \leq D_\text{KL}(\pi_\mu ||\pi_\text{new})$. From this, we derive the following inequality:
> > > > >
> > > > > $\mathbb{E}\_{a_t \sim \pi_\text{new}} [ Q^{\pi_\mu}(s_t, a_t) - \beta \log \pi_\text{new}(a_t|s_t) ] \geq V^{\pi_\mu}(s_t)$ (1)
> > > > >
> > > > > where $V^{\pi_\mu}(s_t) = \mathbb{E}\_{a\sim \pi_\mu(a_t|s_t)} [Q^{\pi_\mu}(s_t, a_t) -\beta \log \pi_\mu (a_t|s_t)]$, which is satisfied in maximum entropy reinforcement learning setting (SAC [1]).
> > > > >
> > > > > Hence:
> > > > >
> > > > > $Q^{\pi_\mu}(s_t, a_t) = r(s_t|a_t) + \beta \log\pi_\text{ref}(a_t|s_t) + V^{\pi_\mu}(s_{t+1})$
> > > > >
> > > > > $\leq r(s_t|a_t) + \beta \log\pi_\text{ref}(a_t|s_t) + \mathbb{E}\_{a_{t+1}\sim \pi_\text{new}}[Q^{\pi_\mu}(s_{t+1}, a_{t+1}) - \beta \log \pi_\text{new} (a_{t+1}|s_{t+1})]$
> > > > >
> > > > > $\cdots$
> > > > >
> > > > > $\leq Q^{\pi_\text{new}}(s_t, a_t)$
> > > > >
> > > > > where we iteratively expand $Q^{\pi_\mu}$ by applying the soft Bellman equation and the bound (1).
> > > > >
> > > > > Since $\pi_\text{new}$ is an improved policy over $\pi_\mu$ (i.e., $Q^{\pi_\text{new}(s_t, a_t)} \geq Q^{\pi_\mu}(s_t, a_t)$), and $\pi_\theta$ is trained to closely match $\pi_\text{new}$, it follows that $\pi_\theta$ will also be an improved policy over $\pi_\mu$ provided the approximation is accurate. Formally, we have:
> > > > >
> > > > > $V^{\pi_\theta}(s) \approx V^{\pi_\text{new}(s)} \geq V^{\pi_\mu}(s)$
> > > > >
> > > > > Since we only incorporate the correct trajectory from the historical data, we can assert that policy \(\pi_\mu\) will not perform worse than the old policy $\pi_\text{old}$. Therefore, we have the relationship:
> > > > >
> > > > > $V^{\pi_\theta}(s) \approx V^{\pi_\text{new}}(s) \geq V^{\pi_\mu}(s) \geq V^{\pi_\text{old}}(s)$
> > > > >
> > > > > The policy improvement process is well-established. Therefore, **using replay buffer for training $V$ and $A$ is feasible**.
> > > > >
> > > > > If the data in the replay buffer is not utilized, the same conclusion holds. However, in this case, $V^{\pi_\mu}$ and $Q^{\pi_\mu}$ are replaced by $V^{\pi_\text{old}}$ and $Q^{\pi_\text{old}}$.
> > > > >
> > > > > **We experimentally confirm that utilizing $V^{\pi_\mu}$ can achieve better performance improvement compared to $V^{\pi_\text{old}}$**. The results are presented in Figure 8.
> > > > >
> > > > > Kindly refer to Appendix A.2 for a detailed proof.

---

> > > > > > ### Author Response · Authors · 2024-11-24
> > > > > > **Response to Reviewer gjpa (Part 2 of 2)**
> > > > > >
> > > > > > ## Response to Question2: Reproducibility
> > > > > >
> > > > > > **1\. Randomness**
> > > > > >
> > > > > > Thank you for your valuable suggestion. We conduct four runs of all baseline models as well as DigiRL with curriculum learning (CL) and present the resulting error bars in Figure 13. The results indicate that WebRL consistently outperforms all baseline methods and DigiRL with CL, demonstrating its effectiveness.
> > > > > >
> > > > > > **2\. Incomplete code**
> > > > > >
> > > > > > Thank you for your reminder. We improve the code repository and include all the scripts, including:
> > > > > >
> > > > > > - Script for SFT training
> > > > > > - Script for WebRL training
> > > > > > - Script for task generation
> > > > > > - Script for interaction
> > > > > >
> > > > > > After completing the SFT training process, each subsequent curriculum learning phase involves the following steps in sequence: task generation, interaction, and WebRL training.
> > > > > >
> > > > > > WebRL is trained using models with over 8 billion parameters, which requires a distributed training approach.
> > > > > > Meanwhile, the interaction process is also complex, involving environment deployment, model deployment, and data processing. **It is challenging to integrate these interaction steps directly into the distributed training script**.
> > > > > >
> > > > > > Therefore, the interaction process and the training process are executed separately.
> > > > > >
> > > > > > ----
> > > > > >
> > > > > > We appreciate the reviewer's suggestion and believe this additional analysis strengthens the validity of our work.
> > > > > >
> > > > > > If you have any further questions or concerns about our paper, please feel free to let us know. If our response has addressed your concerns, we would be truly grateful if you could kindly consider the possibility of revising our score. Thank you for your time and understanding.
> > > > > >
> > > > > > Best wishes,
> > > > > >
> > > > > > All authors
> > > > > >
> > > > > > ---
> > > > > >
> > > > > > [1] Soft Actor-Critic: Off-Policy Maximum Entropy Deep Reinforcement Learning with a Stochastic Actor. ICML 2018

---

> > > > > > > ### Author Response · Authors · 2024-11-25
> > > > > > > **A Friendly Reminder**
> > > > > > >
> > > > > > > Dear Reviewer gjpa,
> > > > > > >
> > > > > > > We sincerely appreciate the time and effort you have dedicated to reviewing our paper. As the discussion period is approaching its conclusion, we kindly wish to remind you that there are two days remaining for any additional comments or questions.
> > > > > > >
> > > > > > > We would be most grateful for the opportunity to address any further concerns or feedback you might have before the discussion period ends.
> > > > > > >
> > > > > > > Thank you once again for your valuable insights and support.
> > > > > > >
> > > > > > > Best regards,
> > > > > > >
> > > > > > > Authors

---

> > > > > > > ### Comment · Reviewer_gjpa · 2024-11-25
> > > > > > >
> > > > > > > Thanks for the clarification. The implementation raised my great concern. If not mentioned explicitly, readers would assume WebRL is an end-to-end method, where the whole training process can be done by running one script once. It seems that in each phrase, WebRL needs to manually run three scripts one by one. Users need to wait for the completion of the previous script and then run the next script. Moreover, WebRL needs to be trained with 10 phrases. The whole training process is overly torturous and with poor reproducibility, which should be at least mentioned in the limitations. I believe the current implementation will not contribute to the community. I urge the authors to clear up the code and propose an end-to-end pipeline.

---

> > > > > > > > ### Author Response · Authors · 2024-11-29
> > > > > > > > **Response to Reviewer gjpa**
> > > > > > > >
> > > > > > > > Dear Reviewer gjpa,
> > > > > > > >
> > > > > > > > We apologize for any concern caused by our code implementation. To clarify, **WebRL is an end-to-end method** that enables the entire training process to run autonomously, without requiring any human intervention. WebRL can be executed within a single script.
> > > > > > > >
> > > > > > > > We appreciate your valuable suggestion, and in response, **we add an end-to-end execution script in the code repository**. This script enables a streamlined and seamless execution of WebRL's complete training process.
> > > > > > > >
> > > > > > > > - - -
> > > > > > > >
> > > > > > > > We appreciate the reviewer's suggestion and believe this additional analysis strengthens the validity of our work.
> > > > > > > >
> > > > > > > > If you have any further questions or concerns about our paper, please feel free to let us know. If our response has addressed your concerns, we would be truly grateful if you could kindly consider the possibility of revising our score. Thank you for your time and understanding.
> > > > > > > >
> > > > > > > > Best wishes,
> > > > > > > >
> > > > > > > > All authors

---

> > > > > > > > > ### Comment · Reviewer_gjpa · 2024-12-01
> > > > > > > > >
> > > > > > > > > After checking the latest implementation, I found that the quality had a significant improvement, so I raised my score accordingly. I urge the authors to further polish the implementation for better reproducibility and to be user-friendly in the camera-ready version.

---

> > > > > > > > > > ### Author Response · Authors · 2024-12-02
> > > > > > > > > > **Response to Reviewer gjpa**
> > > > > > > > > >
> > > > > > > > > > Dear Reviewer gjpa,
> > > > > > > > > >
> > > > > > > > > > Thank you very much for your kind acknowledgment and for taking the time to review our revised manuscript and code implementation! We sincerely appreciate your thoughtful score adjustment. We will continue working to further enhance our code implementation based on your valuable feedback.
> > > > > > > > > >
> > > > > > > > > > Once again, we sincerely thank you for your invaluable feedback and support, which have significantly enhanced our work.
> > > > > > > > > >
> > > > > > > > > > Best regards,
> > > > > > > > > >
> > > > > > > > > > Authors

---

### Official Review · Reviewer_33ZD · 2024-10-31

**Soundness:** 3
**Presentation:** 4
**Contribution:** 3
**Rating:** 8
**Confidence:** 4

**Summary:**

Authors present WebRL, a new framework to fine-tune LLMs for interactive web tasks with sparse rewards. WebRL leverages a curriculum learning setup to perform online actor-critic RL over generated task sets of increasing difficulty. The completion of these generated tasks is assessed autonomously using a learned reward model. Experimental studies are conducted on WebArena-Lite. There, authors showcase the performance advantages of their approach wrt prompt-based method and learning-based methods such as imitation learning or reinforcement learning.

**Strengths:**

This paper tackles a very important topic.
It is well written, I enjoyed reading it.
Authors showcase an impressive performance gap over considered baselines.
Authors present an in-depth analysis of their approach, which is valuable

**Weaknesses:**

### Comparison to no-curriculum RL

The no-curriculum reinforcement learning baseline considered in this paper is Digi-RL. It is in my opinion the most important baseline. It would be good to better understand the differences between Digi-RL and WebRL. Is WebRL  == "Digi-RL with a Curriculum Learning (CL) setup, a KL loss and a replay buffer" ? If no, what are the differences ? What I mean behind this question is that ideally to propose a new CL method it is better to plug the CL method on a well known learner, rather than proposing a new learner AND a curriculum learning method.

### Reliance on human-filtering for curriculum steps

An important weakness is that the curriculum learning process is not autonomous: generated new tasks are ultimately curated by a human. It does not mean the approach is not interesting, but it might be good to emphasize a bit more this limitation, which is only mentioned on line l.261

### Decision

Despite these issues I believe the paper is interesting enough to be accepted. I will go for a weak accept and look forward to the discussion to update my score.


### minor/typos

Be careful about citing journals/conferences and not arxiv, i.e. "Agentbench: Evaluating llms as agents" was published at ICLR 2024. Many arxiv citations in the current version

"(Rawles et al., 2024;Yang et al., 2023; Xie et al., 2024)." --> always cite from oldest to newest

l.162 "progressive" --> progressively

l257 "Detailed prompts are provided in the Appendix § A.3." --> Appendix A.3 is empty of text, although I managed to find the figures. Make sure to reference and briefly discuss each figure in the appendix text.0

**Questions:**

See weaknesses and the question about Digi-RL.

---

> ### Author Response · Authors · 2024-11-18
> **Response to Reviewer 33ZD**
>
> Dear Reviewer 33ZD,
>
> Thank you for your valuable feedback. We deeply appreciate your insights and are fully committed to addressing these points to enhance the quality of our work.
>
> ## Response to Weakness1: Comparison to no-curriculum RL
>
> **1\. The similarity and difference between WebRL and DigiRL**
>
> Similarity:
>
> Both WebRL and DigiRL utilize off-policy RL algorithms and employ the replay buffer.
>
> Difference:
>
> - **Curriculum Learning**. WebRL incorporates curriculum learning, creating tasks tailored to the agent's current capabilities at each phase. DigiRL, however, operates on a pre-defined set of tasks.
> - **Data contained in Replay Buffer**. WebRL's replay buffer stores only successful task trajectories, while DigiRL includes all historical trajectories, regardless of success. WebRL excludes error trajectories because the Generalized Advantage Estimation (GAE) method we use considers the final reward of a trajectory to estimate state advantages. If a step is correct but subsequent errors lead to task failure, that step receives a negative final reward. This distorts advantage estimation and complicates data reuse. To avoid this issue, WebRL discards such trajectories.
> - **Replay Buffer Sampling**. WebRL filters historical experiences using perplexity (PPL) when sampling from the replay buffer, while DigiRL samples experiences randomly.
> - **Policy Update Algorithm**. Although both use off-policy RL algorithms, WebRL's optimization objective includes KL constraints with a reference policy. In contrast, DigiRL's AWR optimization lacks KL constraints. Further details can be found in the analysis below.
>
> **2\. The Need for New Learning Algorithm**
>
> We outline the differences between our algorithm and PPO, AWR (used in DigiRL), and DPO in Appendix A.2. Below, we offer additional clarification.
>
> (1) Theoretical analysis:
> - **Comparison with PPO**: PPO is an on-policy algorithm, whereas our approach is off-policy. In web scenarios, including WebArena or real-world web pages, interaction is expensive and inefficient, making on-policy algorithms unsuitable.
> - **Comparison with DPO**: While both are off-policy, DPO requires constructing pairwise data. In WebAgent scenarios, creating such data is challenging because implementing state backtracking on web pages is difficult
> - **Comparison with AWR**: AWR is also off-policy. However, it does not incorporate a KL constraint with the reference policy and cannot effectively reduce the likelihood of actions with negative advantages.
>
> (2) Experimental verification:
>
> We compare the settings of DigiRL plus curriculum learning with WebRL, as presented in Figure 11. The results indicate that, while the performance of DigiRL plus curriculum learning improves compared to the vanilla version, it remains less effective than WebRL.
>
> ## Response to Weakness2: Reliance on human-filtering for curriculum steps
>
> Thank you for bringing this to our attention. We sincerely apologize for the confusion our previous expression may have caused. Here’s a clearer explanation of the manual check process:
>
> Initially, we conduct a manual review of the generated instructions and identify instructions that could not be completed in WebArena. Based on these observations, we develop a set of rules to filter out such instructions. These rules are then integrated into the prompt used by GPT-4o, enabling it to automatically exclude instructions that are not feasible in WebArena. Then, throughout the curriculum learning process, we use GPT-4o with this prompt to filter out infeasible instructions.
>
> For further clarity, we add this filter prompt to Figure 15 (Appendix D).  In a test of 100 instructions, GPT-4o with this prompt demonstrates 100% recall and 99% precision in identifying tasks deemed infeasible within WebArena.
>
> Additionally, this check process is essential because WebArena is a simulation environment with limitations compared to real websites. For instance, on OSS website, real shopping platforms allow modifications to delivery addresses, while WebArena does not. When using GPT to generate new tasks, it may inadvertently create tasks feasible on real websites but impossible in WebArena. Using unfeasible instructions as seeds for generating new ones could lead to an increasing proportion of unachievable tasks.
>
> ## Response to Weakness3: minor/typos
>
> Thank you for your detailed review! We have addressed the citation issues in the updated manuscript. Additionally, we have included descriptions that refer to and discuss each figure of detailed prompts (Appendix D).
>
> ---
>
> We appreciate the reviewer's suggestion and believe this additional analysis strengthens the validity of our work.
>
> If you have any further questions or concerns about our paper, please feel free to let us know. If our response has addressed your concerns, we would greatly appreciate it if you could kindly consider the possibility of revising our score. Thank you for your time and understanding.
>
> Best wishes,
>
> All authors

---

> ### Author Response · Authors · 2024-11-25
> **A Friendly Reminder**
>
> Dear Reviewer 33ZD,
>
> We sincerely appreciate the time and effort you have dedicated to reviewing our paper. As the discussion period is approaching its conclusion, we kindly wish to remind you that there are two days remaining for any additional comments or questions.
>
> We would be most grateful for the opportunity to address any further concerns or feedback you might have before the discussion period ends.
>
> Thank you once again for your valuable insights and support.
>
> Best regards,
>
> Authors

---

> ### Comment · Reviewer_33ZD · 2024-11-25
> **Answer**
>
> Authors addressed my main concerns. I will increase my score to 8, this is a good paper, with open source code.
>
> I suggest clearly mentioning the appendix/fig 12 experiment, which is important to understand that you have 2 contributions: the curriculum learning strategy and an off-policy learning method well suited in your setting.
>
> I now understand that the human reliance for the curriculum is not strong: it is just expert knowledge over the domain to setup the LLM-based task generator.

---

> > ### Author Response · Authors · 2024-11-26
> >
> > Dear Reviewer 33ZD,
> >
> > Thank you very much for your kind acknowledgment and for taking the time to review our revised manuscript! We are delighted to hear that you are satisfied with our revisions and that your concerns have been addressed. We also sincerely appreciate your thoughtful score adjustment.
> >
> > Following your suggestion, we explicitly reference the appendix/fig 12 experiment in the Ablation Study section to improve clarity.
> >
> > Once again, we sincerely thank you for your invaluable feedback and support, which have significantly enhanced our work.
> >
> > Best regards,
> >
> > Authors

---

### Official Review · Reviewer_8quU · 2024-11-03

**Soundness:** 2
**Presentation:** 2
**Contribution:** 3
**Rating:** 6
**Confidence:** 4

**Summary:**

This paper proposes WebRL: a framework for training agents that can navigate the web using reinforcement learning. They apply their framework to train Llama-3 8B and Llama-3 70B agents that navigate the web, with strong performance on WebArena-Lite.

The overall process for training the agents is:

1. Run the agent on an initial set of instructions from the WebArena-Lite training set, and score them with the WebArena-Lite reward function. Initialize the replay buffer with successful trajectories. Initialize the failure set with instructions from the failing trajectories. Train an outcome-supervised generative reward model (ORM) using these trajectories to predict whether a task has been successfully completed.
2. Generate 500 new instructions using GPT-4o, prompting it to create instructions similar to the ones from the failure set, but more diverse.
3. Filter the generated instructions. First filter them using the trained critic (value function), keeping instructions which the critic predicts are difficult but possible (between 0.05 and 0.75). Then filter through manual human review, to eliminate bad instructions.
4. Run the agent on the instructions to generate a set of trajectories to generate a training set. Extend the training set with actions from the replay buffer whose perplexity is between 1/0.95 and 1/0.5.
5. Update the agent policy on these trajectories using an RL algorithm that combines policy gradients, advantage estimation, and replay buffers.
6. Add successful trajectories to the replay buffer, and add instructions from failing trajectories to the failure set.
7. Go back to step 2 to begin a new phase.

The authors compare against a variety of baselines, including imitation learning and DigiRL, as well as a variety of ablations of their method.

**Strengths:**

* Training LLMs to navigate the web is an important area of research
* The results in the paper appear to be quite strong.
* The experimental results involve a good variety of baselines and ablations.

**Weaknesses:**

## RL algorithm

There are a few weaknesses with the RL approach (Section 2.1):

* It’s not clear to me why the authors derive a new RL algorithm. I don’t see what benefits this would have over the approach of using PPO with the KL-regularized reward (as in [Ziegler et al](https://arxiv.org/abs/1909.08593)). In fact, I believe that the authors’ algorithm is nearly equivalent to REINFORCE with a value function baseline – the only difference I see is that in the authors’ algorithm, the gradient is multiplied by a factor of $\beta$, which seems undesirable to me (ideally the scale of the gradient wouldn’t be sensitive to $\beta$).
* In addition, this new algorithm is an on-policy algorithm – but it is used alongside a replay buffer, which is naturally off-policy. Presumably an off-policy correction term would be needed.
* Also, the math used to derive the new algorithm is flawed. Specifically, to derive Equation (5) from Equation (4), the authors rely on the result that the optimal policy in an RL environment is unaffected by potential shaping. However, this result does not hold in the KL-regularized RL setting, so this step is invalid.

## Presentation

I found the paper somewhat hard to understand – it was hard to tell what the various components were and how they fit together. Figure 1 is quite helpful in clarifying this with an architecture diagram, but I would recommend that the authors also include an algorithm description, similarly to the outline provided in my summary above. (Please do also let me know if my summary is wrong.)

It is somewhat misleading to say that the task distribution is “self-evolving” when it involves a manual step (L261):

> Then, we manually check the instructions to eliminate instructions that are clearly unreasonable or impossible to accomplish.

## Other

**Single benchmark.** Since the authors only evaluate on a single task (WebArena-Lite) it is hard to tell which aspects of their setup are specific to WebArena and which are generalizable to any LLM control environment. Given the importance of browser control, and the diversity of tasks within WebArena, this is only a minor weakness. However, it could be addressed by applying the WebRL recipe to similar settings, such as Android (in which DigiRL was evaluated).

**Task complexity results.** On lines 392-396 the authors say:

> Our results show that WEBRL performs well across different complexity levels, particularly excelling in more complex instructions. In contrast, while DigiRL uses online learning, it struggles with higher complexity due to its focus on a fixed set of tasks, limiting its adaptability.

However, this doesn’t seem to be supported by Figure 5, where it appears that both WebRL and DigiRL see significantly lower task success rates at higher complexity levels.

**KL ablation.** The authors say they consider a version of WebRL without the KL regularization – however, they say that this involves running SFT. This seems like the wrong way to do it – instead, the policy should be trained with an RL algorithm, but without the KL regularization. This would produce a similar gradient as SFT, except that it would be weighted by the advantage function. The baseline reported in the paper is also interesting, though it should be called “WebRL-SFT” or something similar rather than “WebRL w/out KL”.

**Questions:**

* Why do you develop a new RL approach, rather than applying PPO (or a similar algorithm) to the KL-regularized reward?
* Why are the success rates in Figure 5 so low? Shouldn’t they be around 40% on average for WebRL, to be consistent with Table 1? Perhaps the denominator in Figure 5 was accidentally set to “all tasks” rather than tasks at the appropriate complexity level.

---

> ### Author Response · Authors · 2024-11-18
> **Response to Reviewer 8quU (Part 1 of 2)**
>
> Dear Reviewer 8quU,
>
> Thank you for your valuable feedback. We truly value your insights and are fully committed to addressing these points to enhance the quality of our work.
>
> ## Response to Weakness1: Questions of RL algorithm
>
> We provide a more detailed derivation of our algorithm in Appendix A.1, where we also explain why it operates in an off-policy manner. Additionally, we present a detailed comparison of our algorithm with other algorithms, including PPO, DPO, and AWR, in Appendix A.2. For convenience, the key distinctions are summarized concisely in Table 4.
>
> Regarding your three questions:
>
> **1\. Why do the authors derive a new RL algorithm**
>
> - **Our algorithm is an off-policy approach, unlike PPO, which is on-policy.** In web scenarios, including WebArena or real-world web pages, interaction is expensive and inefficient, making on-policy algorithms unsuitable. To address this limitation, we propose a novel off-policy algorithm to enhance efficiency.
> - **Difference from REINFORCE with a value function baseline.** The gradient of our algorithm is: $-2\beta \mathbb{E}_{\nu} [(A^*(s, a, I) - \beta \log\frac{\pi\_\theta(a|s, I)}{\pi\_{\text{ref}}(a|s, I)}) \nabla\_\theta \log\pi\_\theta(a|s,I)]]$, while the gradient of REINFORCE with a value function baseline is $-\mathbb{E}\_{\nu} [(A(s, a, I) \nabla\_\theta \log\pi\_\theta(a|s,I) ]$. Our gradient includes a KL divergence term which is derived from the optimization objective in Eq. 1, rather than being arbitrarily added.  Meanwhile, the derivation process of the loss function of our algorithm and REINFORCE with a value function baseline is completely different.
> - **$\beta$ coefficient in the gradient.** While our algorithm will generate beta coefficients in the gradient, DPO will also produce beta coefficients in its gradient. The $\beta$ coefficient will not impact the feasibility of our algorithm.
>
> **2\. On-policy algorithm**
>
> In Appendix A.1, we provide a detailed explanation of why our algorithm is off-policy (lines 784-788). Briefly, we first derive the optimal solution $\pi^*$ for the objective function (Eq. 1). Then we build the optimization target of $\pi\_\theta$:
> ${\arg\min}\_{\theta}\  \mathbb{E}\_{\nu} [ ( \log\pi\_\theta(a|s, I) - \log\pi^*(a|s, I) )^2 ]$.
>
> This optimization target leads to the loss function:
>
> $\mathcal{L}(\pi\_\theta) = \mathbb{E}\_{\nu} [ ( \beta \log \frac{\pi\_\theta(a|s, I)}{\pi\_{\text{ref}}(a|s, I)} - A^*(s, a, I) )^2 ]$
>
> For any given state-action pair in $\nu$, $\pi_\theta$ is expected to match the target policy $\pi^*$. There is no restriction on the data distribution $\nu$, indicating the algorithm can function in an off-policy setting.
>
> **3\. The math used to derive the new algorithm is flawed**
>
> You are correct that reward shaping does not apply under the KL-regularized RL setting. Upon further inspection, we find that Eq. 4 is redundant due to an oversight on our part.  In fact, Eq. 5 can be directly derived from Eq. 2 and Eq. 3 without relying on Eq. 4.
> By substituting Eq. 3 into Eq. 4, we can derive Eq. 5. The derivation process is presented below:
>
> Eq. 3: $\pi^*(a_t | s_t, I) = e^{(Q^*(s_t, a_t, I) - V^*(s_t, I))/\beta}$
>
> Eq. 4: $Q^*(s_t, a_t, I) = r(s_t, a_t, I) + \beta \log \pi_{\text{ref}}(a_t | s_t, I) + V^*(s_{t+1}, I)$
>
> Substituting Eq. 3 into Eq. 4: $\pi^*(a_t | s_t, I) = e^{\left(Q^*(s_t, a_t, I) - V^*(s_t, I)\right)/\beta} = e^{(\beta \log \pi_{\text{ref}}(a_t | s_t, I) + r(s_t, a_t, I) + V^*(s_{t+1}, I) - V^*(s_t, I))/\beta}$
>
> $\rightarrow \beta \log \pi^*(a_t|s_t, I)=\beta \log \pi_{\text{ref}}(a_t | s_t, I) + r(s_t, a_t, I) + V^*(s_{t+1}, I) - V^*(s_t, I)$
>
> $\rightarrow \beta \log \frac{\pi^*(a_t|s_t, I)}{\pi_{\text{ref}}(a_t|s_t, I)} = r(s_t, a_t, I) + V^*(s_{t+1}, I) - V^*(s_t, I) = A^*(s_t, a_t, I)$
>
> As a result, the derivation of Eq. 5 does not depend on reward shaping, which alleviates concerns regarding the validity of the formula itself. We sincerely apologize for any confusion caused by this oversight and greatly appreciate your understanding.

---

> ### Author Response · Authors · 2024-11-18
> **Response to Reviewer 8quU (Part 2 of 2)**
>
> ## Response to Weakness2: Presentation
>
> **1\. Components of WebRL**
>
> Thank you for your suggestion! We add the pseudocode for WebRL in Algorithm 1 (lines 972-998), which provides a detailed description of the entire WebRL training process.
>
> Your summary of the process is generally accurate. However, I’d like to clarify that the ORM involves a distinct training process. To build the training data of ORM, we first enhance the WebArena-Lite dataset by rewriting its instructions. Next, we collect rollouts from all baseline methods using this augmented dataset. The rollouts, combined with the original WebArena-Lite training samples, form the final training dataset of 12,200 samples. We provide additional training details of ORM in Appendix B.3.
>
> **2\. The manual step in "self-evolving"**
>
> Thank you for bringing this to our attention. We sincerely apologize for the confusion our previous expression may have caused. Here’s a clearer explanation of the manual check process:
>
> Initially, we conduct a manual review of the generated instructions and identify instructions that could not be completed in WebArena. Based on these observations, we develop a set of rules to filter out such instructions. These rules are then integrated into the prompt used by GPT-4o, enabling it to automatically exclude instructions that are not feasible in WebArena. Then, throughout the curriculum learning process, we use GPT-4o with this prompt to filter out infeasible instructions.
>
> For further clarity, we add this filter prompt to Figure 15 (Appendix D).  In a test of 100 instructions, GPT-4o with this prompt demonstrates 100% recall and 99% precision in identifying tasks deemed infeasible within WebArena.
>
> Additionally, this check process is essential because WebArena is a simulation environment with limitations compared to real websites. For instance, on OSS website, real shopping platforms allow modifications to delivery addresses, while WebArena does not. When using GPT to generate new tasks, it may inadvertently create tasks feasible on real websites but impossible in WebArena. Using unfeasible instructions as seeds for generating new ones could lead to an increasing proportion of unachievable tasks.
>
> ## Response to Weakness3: Other
>
> **1\. Single benchmark**
>
> Thank you for your suggestion. Given the complexity of configuring the Android environment in DigiRL, we plan to explore using WebRL for Android at a later stage.
>
> **2\. Task complexity results**
>
> We apologize for the oversight in Figure 5.  Previously, we incorrectly calculated accuracy across all complexity levels by dividing by the total number of tasks (165) instead of the number of tasks at each level. We have corrected Figure 5 in the new manuscript. The corrected Figure 5 shows that WebRL achieves over 34% accuracy across tasks of varying complexity levels. Notably, at complexity level 4, the baseline method's maximum accuracy is only 18%.
>
> **3\. KL ablation**
>
> Thank you for your suggestion! We replace WebRL w/o KL with REINFORCE using a value function baseline and include the updated results in the new manuscript (Figure 3). The updated results demonstrate that REINFORCE with a value function baseline outperforms the previous SFT results, regardless of whether a replay buffer is used. However, it still performs weaker than the corresponding WebRL counterparts.
>
> ## Response to Question1: Algorithm Advantage
>
> We provide a detailed comparison of our algorithm with PPO, DPO, and AWR in Appendix A.2, with a summary in Table 4. Key differences are outlined below:
>
> - **Comparison with PPO**: PPO is an on-policy algorithm, whereas our approach is off-policy.
> - **Comparison with DPO**: While both are off-policy, DPO requires constructing pairwise data. In WebAgent scenarios, creating such data is challenging because implementing state backtracking on web pages is difficult, making it hard to collect outcomes of different actions on the same page state.
> - **Comparison with AWR**: AWR is also off-policy. However, it does not incorporate a KL constraint with the reference policy and cannot effectively reduce the likelihood of actions with negative advantages.
>
> ## Response to Question2: Denominator error in Figure 5
>
> Kindly refer to Point 2 "Task complexity results" under "Response to Weakness 3: Other"
>
> ---
>
> We appreciate the reviewer's suggestion and believe this additional analysis strengthens the validity of our work.
>
> If you have any further questions or concerns about our paper, please feel free to let us know. If our response has addressed your concerns, we would greatly appreciate it if you could kindly consider the possibility of revising our score. Thank you for your time and understanding.
>
> Best wishes,
>
> All authors

---

> ### Comment · Reviewer_8quU · 2024-11-19
> **Still not convinced it is a valid off-policy algorithm**
>
> I have read the author response and the other reviews and am maintaining my score. I would have gone up a point to 7 on a 10-point scale, but I wouldn't go up to 8, which is the next rating available in ICLR -- maybe this should be considered a "strong 6".
>
> Thanks for the updates, which in my view have made the paper stronger. In particular, I especially appreciate the addition of Algorithm 1, which makes it clearer how the method works (e.g. I hadn't realized that in each phase you keep prompting GPT-4o for more instructions until you achieve the required number of instructions), and the hyperparameters used for training, which should enhance reproducibility. (Incidentally, what's the value of K, the number of instructions per phase? I didn't see it immediately but may have missed it.) The updated derivation of the RL algorithm is also better; I agree it fixes the mistake I pointed out about potential shaping not being applicable to this setting.
>
> I'm still not convinced that the derived RL algorithm is an appropriate off-policy algorithm. I agree if you could directly optimize the loss in Equation (5) of $$E\bigg[\bigg(\beta \log \frac{\pi_{\theta}(a \mid s, I)}{\pi_{\text{ref}}(a \mid s, I)} - A^*(s, a, I)\bigg)^2\bigg]$$ then that would be a valid off policy algorithm. However, we do not have access to $A^*$, the advantage of the optimal policy. Instead, you're using an advantage estimator which is based on a value network that is trained on data collected throughout training -- this will usually be an estimate of the advantage of the policy being learned, that is $A(s, a, I)$, rather than the advantage of the optimal policy, $A^*(s, a, I)$, though the introduction of a replay buffer complicates the matter.
>
> I also agree with the other reviewers that the paper could be improved scientifically through more careful ablations (I would also add qualitative analysis of failures), and generally by adding and testing each new component separately, rather than creating an entire new system from scratch. Some design decisions whose impact is unknown: the decision to only keep successful trajectories in the replay buffer, the filtering based on action perplexity, the prompting strategy for generating new instructions, ...

---

> > ### Author Response · Authors · 2024-11-24
> > **Response to Reviewer 8quU (Part 1 of 2)**
> >
> > Dear Reviewer 8quU,
> >
> > Thank you for your valuable suggestions. We sincerely appreciate your insights. We arrange our responses in the order your questions are raised, and hope they clarify your concerns.
> >
> > ## Response to Question1: Number of instructions per phase
> >
> > The number of instructions per phase is set at 500, as outlined in lines 978–979.
> >
> > ## Response to Question2: Replacing $A^*(s, a, I)$ with $A(s, a, I)$
> >
> > Thank you for bringing this to our attention. We address this in the updated manuscript, where we provide a detailed proof of the feasibility of using $A(s, a, I)$ in Appendix A.2. Briefly, **we demonstrate that using $A$ to train the policy can lead to policy improvement.**
> >
> > The current policy is denoted by $\pi_\text{old}$. The training dataset $D$ is composed of trajectories sampled using $\pi_\text{old}$ as well as historical experiences stored in a replay buffer. The behavioral policy associated with the dataset $D$ is referred to as $\pi_{\mu}$. The corresponding value function and action-value function under $\pi_{\mu}$ are denoted as $V^{\pi_{\mu}}$ and $Q^{\pi_{\mu}}$,  respectively.
> >
> > **We demonstrate that minimizing the objective $\arg\min_\theta \mathbb{E}\_D[(\log_{\pi_\theta}(a_t|s_t) - \exp (\frac{Q^{\pi_\mu}(s_t, a_t) - V^{\pi_\mu}(s_t)}{\beta}))^2]$ can lead to policy improvement.**
> >
> > The new policy $\pi_\text{new}(a_t|s_t)$ is defined as:
> > $\pi_\text{new}(a_t|s_t) = \exp\left(\frac{Q^{\pi_\mu}(s_t, a_t) - V^{\pi_\mu}(s_t)}{\beta}\right)$
> >
> > It can be shown that:  $D_\text{KL}(\pi_\text{new}||\pi_\text{new})
> > \leq D_\text{KL}(\pi_\mu ||\pi_\text{new})$. From this, we derive the following inequality:
> >
> > $\mathbb{E}\_{a_t \sim \pi_\text{new}} [ Q^{\pi_\mu}(s_t, a_t) - \beta \log \pi_\text{new}(a_t|s_t) ] \geq V^{\pi_\mu}(s_t)$ (1)
> >
> > where $V^{\pi_\mu}(s_t) = \mathbb{E}\_{a\sim \pi_\mu(a_t|s_t)} [Q^{\pi_\mu}(s_t, a_t) -\beta \log \pi_\mu (a_t|s_t)]$, which is satisfied in maximum entropy reinforcement learning setting (SAC [1]).
> >
> > Hence:
> >
> > $Q^{\pi_\mu}(s_t, a_t) = r(s_t|a_t) + \beta \log\pi_\text{ref}(a_t|s_t) + V^{\pi_\mu}(s_{t+1})$
> >
> > $\leq r(s_t|a_t) + \beta \log\pi_\text{ref}(a_t|s_t) + \mathbb{E}\_{a_{t+1}\sim \pi_\text{new}}[Q^{\pi_\mu}(s_{t+1}, a_{t+1}) - \beta \log \pi_\text{new} (a_{t+1}|s_{t+1})]$
> >
> > $\cdots$
> >
> > $\leq Q^{\pi_\text{new}}(s_t, a_t)$
> >
> > where we iteratively expand $Q^{\pi_\mu}$ by applying the soft Bellman equation and the bound (1).
> >
> > Since $\pi_\text{new}$ is an improved policy over $\pi_\mu$ (i.e., $Q^{\pi_\text{new}(s_t, a_t)} \geq Q^{\pi_\mu}(s_t, a_t)$), and $\pi_\theta$ is trained to closely match $\pi_\text{new}$, it follows that $\pi_\theta$ will also be an improved policy over $\pi_\mu$ provided the approximation is accurate. Formally, we have:
> >
> > $V^{\pi_\theta}(s) \approx V^{\pi_\text{new}(s)} \geq V^{\pi_\mu}(s)$
> >
> > Since we only incorporate the correct trajectory from the historical data, we can assert that policy \(\pi_\mu\) will not perform worse than the old policy $\pi_\text{old}$. Therefore, we have the relationship:
> >
> > $V^{\pi_\theta}(s) \approx V^{\pi_\text{new}}(s) \geq V^{\pi_\mu}(s) \geq V^{\pi_\text{old}}(s)$
> >
> > The policy improvement process is well-established. By iteratively performing policy evaluation and policy improvement, the policy $\pi_\theta$ can be made to converge toward the optimal policy $\pi^*$.
> >
> > **If the data in the replay buffer is not utilized, the same conclusion holds**. However, in this case, $V^{\pi_\mu}$ and $Q^{\pi_\mu}$ are replaced by $V^{\pi_\text{old}}$ and $Q^{\pi_\text{old}}$.
> >
> > We experimentally confirm that utilizing $V^{\pi_\mu}$ can achieve better performance improvement compared to $V^{\pi_\text{old}}$. The results are presented in Figure 8.
> >
> > Kindly refer to Appendix A.2 for a detailed proof.

---

> > > ### Author Response · Authors · 2024-11-24
> > > **Response to Reviewer 8quU (Part 2 of 2)**
> > >
> > > ## Response to Question3: More ablation studies
> > >
> > > Thank you for your suggestion! The ablation study of the `filtering based on action perplexity` is originally included in the manuscript (Table 2). The result demonstrates that training exclusively on data with either low perplexity or high perplexity will degrade the performance. In contrast, selecting data within a moderate perplexity range yields the best results.
> > > Based on your suggestion, we add the following ablation studies:
> > >
> > > **1\. Contain all trajectories in the replay buffer.**
> > >
> > > The result is shown in Figure 14. We compare two configurations of the replay buffer: one containing only past successful trajectories and another including all past trajectories, both successful and failed. The results indicate that including failed trajectories slows down performance improvement and increases volatility.
> > >
> > > This occurs because the Generalized Advantage Estimation (GAE) method we use considers the final reward of a trajectory to estimate advantages. Even if certain actions in a failed trajectory are correct, they will likely receive a negative advantage when the final reward is zero. This reduces the likelihood of selecting these correct actions, hindering policy improvement and leading to suboptimal performance.
> > >
> > > **2\. The prompt for generating new instructions.**
> > >
> > > We introduce two additional versions of the prompt for generating new instructions, as shown in Figures 17 and 18. The performance comparison results for the two versions and the original prompt are presented in Table 7.
> > >
> > > The results demonstrate that when using a variety of prompts to generate tasks, WebRL consistently delivers strong performance, highlighting the stability of WebRL.
> > >
> > > ---
> > >
> > > We appreciate the reviewer's suggestion and believe this additional analysis strengthens the validity of our work.
> > >
> > > If you have any further questions or concerns about our paper, please feel free to let us know. If our response has addressed your concerns, we would be truly grateful if you could kindly consider the possibility of revising our score. Thank you for your time and understanding.
> > >
> > > Best wishes,
> > >
> > > All authors
> > >
> > > ---
> > >
> > > [1] Soft Actor-Critic: Off-Policy Maximum Entropy Deep Reinforcement Learning with a Stochastic Actor. ICML 2018

---

> ### Author Response · Authors · 2024-11-25
> **A Friendly Reminder**
>
> Dear Reviewer 8quU,
>
> We sincerely appreciate the time and effort you have dedicated to reviewing our paper. As the discussion period is approaching its conclusion, we kindly wish to remind you that there are two days remaining for any additional comments or questions.
>
> We would be most grateful for the opportunity to address any further concerns or feedback you might have before the discussion period ends.
>
> Thank you once again for your valuable insights and support.
>
> Best regards,
>
> Authors

---

### Author Response · Authors · 2024-11-18
**General Response**

Dear Reviewers,

We extend our deep gratitude to the three reviewers' time and effort in reviewing our manuscript. In response to their valuable feedback, we have revised the manuscript and submitted an updated version incorporating the following modifications:

**Main Text Revisions:**
1. Removal of Redundant Equation: Delete Eq. 4, which is redundant.
2. Reorganization of Section 2: Move the introduction of self-evolving curriculum learning to precede the introduction of the RL algorithm of WebRL in Section 2. Additionally, a more detailed explanation of the manual checking process within self-evolving curriculum learning is provided.
3. Updates to Figure 3: Add the WebRL w/o CL configuration and modify the policy update algorithm for WebRL w/o KL from SFT to REINFORCE with a value function baseline.
4. Corrections in Figure 5: Fix an error in the denominator for the percentage calculation, replacing the previous value of 165 with the number of tasks corresponding to each complexity level.
5. Fix Citation Error

**Appendix Revisions:**
1. Appendix A. Expand the derivation and analysis of the policy update algorithm of WebRL and provide comparisons with other algorithms.
2. Appendix B.  Add (1) the observation and action space of WebRL. (2) Pseudocode for WebRL. (3) Training details for WebRL, ORM, baseline methods, and their hyperparameters.
3. Appendix C. Enhance quantitative experiments with (1) Result of error bars of WebRL in each phase. (2) Comparative results for WebRL and DigiRL. (3) Result of error bars of WebRL and baselines. (4) Ablation study of trajectories in replay buffer.
4. Appendix D. Add detailed descriptions of the prompts used.
5. Appendix E. Include a case study demonstrating the generated instructions during the curriculum learning process.
6. Appendix F: Qualitative examples

----

**General Clarification**

| Clarification on manual check process in self-evolving curriculum learning

Here’s a clearer explanation of the manual check process:

Initially, we conduct a manual review of the generated instructions and identify instructions that could not be completed in WebArena. Based on these observations, we develop a set of rules to filter out such instructions. These rules are then integrated into the prompt used by GPT-4o, enabling it to automatically exclude instructions that are not feasible in WebArena. Then, throughout the curriculum learning process, we use GPT-4o with this prompt to filter out infeasible instructions.

For further clarity, we add this filter prompt to Figure 15 (Appendix D).  In a test of 100 instructions, GPT-4o with this prompt demonstrates 100% recall and 99% precision in identifying tasks deemed infeasible within WebArena.

Additionally, this check process is essential because WebArena is a simulation environment with limitations compared to real websites. For instance, on OSS website, real shopping platforms allow modifications to delivery addresses, while WebArena does not. When using GPT-4o to generate new tasks, it may inadvertently create tasks feasible on real websites but impossible in WebArena. Using unfeasible instructions as seeds for generating new ones could lead to an increasing proportion of unachievable tasks.

---

### Meta-Review · Area_Chair_KQJL · 2024-12-19

**Metareview:**

(a) This paper introduces WebRL, a new framework for fine-tuning LLMs for interactive web tasks. The authors propose an actor-critic RL method learning with a self-evolving online curriculum that generates new tasks from unsuccessful attempts. This allows WebRL to operate over sets of generated tasks that become increasingly difficult as training progresses. To assess whether the generated tasks are completed, the authors use a trained outcome reward model. The paper focuses on training open-source LLMs as agents for web navigation and claims to achieve impressive performance on WebArena-Lite, outperforming both prompt-based and learning-based methods.

(b) Strengths:
- Strong performance on WebArena-Lite, significantly outperforming existing state-of-the-art web agents, including those based on proprietary LLMs.
- Addressing key challenges: WebRL tackles critical issues in training web agents, such as the scarcity of training tasks, sparse feedback signals, and policy distribution drift in online learning.
- Novel framework: The paper proposes a novel self-evolving online curriculum RL framework, which dynamically adjusts task complexity.

(c) Weaknesses:
- Clarity on algorithm design: Reviewers found the paper somewhat difficult to understand, particularly regarding why to propose a new RL method for off-policy learning.
- Experimental limitations: The reliance on a single benchmark (WebArena-Lite) makes it difficult to assess the generalizability of the findings to other web environments, especially with the manual intervention steps for each web environment.
- Limited ablation studies and training details: Reviewers identified insufficient ablation studies, particularly regarding the curriculum learning strategy and the impact of individual design choices. Additionally, the paper lacked details about the training process for the outcome reward model (ORM).
- Experimental limitations: Using single benchmark (WebArena)

(d) Given the novelty and strong performance of the paper, I think there is a consensus that the strengths outweigh the weaknesses.

**Additional Comments On Reviewer Discussion:**

Please summarize the discussion and changes during the rebuttal period. What were the points raised by the reviewers? How were each of these points addressed by the authors? How did you weigh in each point in your final decision? (Do not mention reviewer names. Use their anon ids instead.)

1. The reviewers found the paper somewhat difficult to understand, particularly regarding the interplay of different components and the algorithm's derivation.
2. They also raised questions regarding the reason to propose a new RL method and the correctness.
3. While the paper emphasizes "self-evolving" curriculum learning, it relies on manual prompting to create a filter for each web environment. This manual step raises concerns about the scalability and true autonomy of the approach.
4. Experimental limitations
5. Limited ablation studies and training details.

1 and 5 were sufficiently addressed. 2 was partially addressed. The reason to do so instead of using existing methods (or at least comparing to) hasn’t been clarified too much. 3 and 4 remain the limitations of this work.

The author responses helped improve the paper from borderline to accept in my final decision.

---

### Decision · Program_Chairs · 2025-01-22

Accept (Poster)